# Analysis of a southern sub-polar short-term ozone variation event using a Millimeter-Wave Radiometer

Pablo Facundo Orte[1], Elian Wolfram[1, 2, 3], Jacobo Salvador[1, 2], Akira Mizuno[4], Nelson Bègue[5], Hassan Bencherif[5, 8], Juan Lucas Bali[1, 2], Raúl D'Elia[1, 2], Andrea Pazmiño[6], Sophie Godin-Beekmann[6], Hirofumi Ohyama[7], Jonathan Quiroga[1]

[1]Centro de Investigaciones en Láseres y Aplicaciones, UNIDEF (CITEDEF-CONICET), UMI-IFAECI-CNRS-3351, Villa Martelli, Buenos Aires, Argentina
[2]Facultad Regional Buenos Aires, Universidad Tecnológica Nacional, Buenos Aires, Argentina
[3]Consejo Nacional de Investigaciones Científicas y Técnicas (CONICET), Buenos Aires, Argentina
[4]Institute for Space-Earth Environmental Research, Nagoya University, Furo-cho, Chikusa-ku, Nagoya 464-8601, Japan
[5]Laboratoire de l'Atmosphère et des Cyclones, UMR 8105 CNRS/Universite/Météo-France, Université de La Réunion, B.P 7151, 97715, Saint-Denis, La Réunion, France
[6]Laboratoire Atmosphère, Milieux, Observations Spatiales (LATMOS), Institut Pierre Simon Laplace, Université Pierre et, Marie Curie, Université Versailles St-Quentin-en-Yvelines, Centre National de la Recherche Scientifique, Paris, France
[7]National Institute for Environmental Studies (NIES), Tsukuba, Japan
[8]University of KwaZulu-Natal, School of Chemistry and Physics, Westville, Durban, South Africa

*Correspondence to*: Pablo Facundo Orte (porte@citedef.gob.ar)

**Abstract.** Subpolar regions in the southern hemisphere are influenced by the Antarctic polar vortex during austral spring, which induces high and short term ozone variability at different altitudes mainly into the stratosphere. This variation may affect considerably the total ozone column changing the harmful UV radiation that reaches the surface.

With the aim to study ozone with high time resolution at different altitudes in subpolar regions, a Millimeter Wave Radiometer (MWR) was installed at the Observatorio Atmosférico de la Patagonia Austral (OAPA), Río Gallegos, Argentina, (51.6º S; 69.3º W) by 2011. This instrument provides ozone profiles with a time resolution of ~1 hour which enables studies of short term ozone mixing ratio variability from 25 to ~70 km in altitude. This work presents the MWR ozone observations between October 2014 and 2015 focusing on an atypical event of the polar vortex and Antarctic ozone hole influence over Río Gallegos detected from the MWR measurements at 27 and 37 km during November of 2014. During the event, the MWR observations at both altitudes show a decrease of ozone followed by a local peak of ozone amount of the order of hours. This local recovery is observed thanks to the high time resolution of the MWR mentioned. The advected potential vorticity (APV) calculated from the high-resolution advection model MIMOSA (Modélisation Isentrope du transport Méso-échelle de l'Ozone Stratosphérique par Advection) was also analysed at two isentropic levels (levels of constant potential temperature) of 675 and 950 K (~27 km and ~37 km of altitude, respectively) to understand and explain the dynamics at both altitudes and correlate the ozone rapid recovery with the passage of a tongue with low PV values over Río Gallegos. In addition, the MWR dataset was compared for first time with measurements obtained from Microwave Limb Sounder (MLS) at individual altitude levels (27 km, 37 km and 65 km) and with the Differential Absorption Lidar (DIAL) installed in OAPA to analyse the correspondence between the MWR and independent instruments. The MWR-MLS

comparison presents a reasonable correlation with a mean bias error of +5%, -11% and -7% at 27 km, 37 km and 65 km, respectively. The MWR-DIAL comparison at 27 km presents also good agreement with a mean bias error of -1%.

## 1 Introduction

Ozone is an atmospheric trace compound, which reaches its absolute maximum concentration in the stratosphere, between 15
and 35 km, where it forms the "ozone layer" (London et al., 1985). It acts as an absorber of harmful solar UVB radiation, protecting the life on Earth (Salby, 1996, Dobson, 1956). Without atmospheric ozone, life would not be possible as we know it today. Although most production takes place in the mesospheric-stratospheric equatorial region due to the higher level of solar radiation, the maximum ozone concentration is observed over the polar region (Salby, 1996). This zonal distribution is explained by the Brewer–Dobson circulation (Brewer, 1949; Dobson, 1956), which transports ozone-rich air masses from the
Equator to the Pole, into the stratosphere.

Nevertheless, since the 70's the ozone layer has suffered a drastic reduction over the Antarctic region inside the polar vortex during the austral spring seasons, known as "Antarctic ozone hole" (AOH) (Chubachi, 1984; Farman et al., 1985). This ozone destruction is the consequence of human emission of components containing chlorine and bromine into the atmosphere, called Ozone Depleting Substances (ODS) (WMO, 2011). The most direct impact of ozone reduction is the
increase of harmful solar UVB radiation over the surface in polar and subpolar regions (Casiccia et al., 2008; Wolfram et al., 2012).

With the aim to reduce ODS and mitigate ozone depletion, the Montreal Protocol was signed in 1987 banning the use of ODS, and a decrease of these substances in the atmosphere was observed (WMO, 2014). However, the lifetime of these compounds in the atmosphere is very long (e.g, 100 years for some of them) (M. Rigby et al., 2013, 2014; WMO, 2014) and
they will remain for decades in the atmosphere, destroying ozone mainly over the Antarctic polar region.

In spite of the fact that massive ozone depletion is produced over the South Pole, ozone reduction at different height levels were also observed in non-polar regions between 1980s and 1990s (WMO, 2014). Kirchhoff et al. (1997; 1997b) reported TOC ranging from 145 DU to 250 DU in Punta Arenas (53.0'S, 70.9'W)), during low-ozone events during September-December of 1992-1995, for which the climatological average is 330-334 DU (maximum reduction of ~56%). A more recent
study reported reduction of 40-45% in TOC over Río Gallegos on October 2008 and November 2009 (Kuttippurath et al. 2010b). ECC (Electro Chemical Cell) ozone-sonde profiles measurements reflect reductions of around 30 to 50% between 15 km to 32 km of altitude in ozone hole condition (inner) respect to normal condition (outer the ozone hole) in Punta Arenas (Kirchhoff et al.,1997). Similar reduction was observed from a Differential Absorption Lidar at Río Gallegos (Wolfram et al., 2006).

Together with the banning in the use of ODS set by the 1987 Montreal Protocol, the general expectation was that the TOC would recover as the amount of ODS decreased in all regions. Recent studies showed a recovery of the stratospheric ozone column during September (statistically significant) and October (statistically insignificant) for the South Polar Region

(Salomon et al., 2016; Weber et al., 2018; Pazmiño et al., 2018). Ground and space-based observations and models have shown an increase of the total ozone since 2000. Nevertheless, this increase is not significant for the period 2000-2013 (WMO, 2014). Ball et al. (2018) extended this period from 1998 to 2016 and concluded that there are non-significant changes in the total amount of ozone from merged ozone datasets.

Recent partial column ozone analysis from satellite ozone composite indicates a decadal increases in the upper stratosphere that is statistically significant (WMO, 2014; Harris et al., 2015; Steinbrecht et al., 2017; Ball et al., 2017; Frith et al., 2017; Ball et al. 2018) attributed in part, to the decline of the ODS. On the other hand, an unexpected decrease of the partial column ozone in lower stratosphere has been suggested, although with a low level of confidence (Nair et al., 2015; Vigouroux et al., 2015; Ball et al., 2018). A global scale study (between 60°N and 60°S) confirmed a significant decrease of
partial column ozone in lower stratosphere at tropical latitudes with high level of significance, accounting a continued and uninterrupted decline in the order of ~2.2 DU between 1998 and 2016 (Ball et al., 2018).

During the austral spring time, the Antarctic polar vortex changes its size and shape and it can reach subpolar regions due mainly to tropospheric-stratospheric dynamical processes. Hence, the Antarctic polar vortex can overpass the continental South American region in subpolar latitudes and this situation may provoke decreases in the TOC content to unusual levels
due to the passage of masses with low ozone amount, as a consequence of the AOH influence (Pazmiño et al., 2005; Wolfram et al., 2008). The passage of the AOH is identified using the TOC threshold of 220 DU, while other considerable reductions in ozone above 220 DU in link to the polar vortex are mentioned as "ozone hole influence". A particular case of unusual persistence of the AOH influence over southern Argentina was observed during November 2009 with satellite and ground-based instruments, which led to an increase in the risk of UVB radiation on the surface (Wolfram et al., 2012). This
phenomenon was first observed by Kirchhoff et al. (1996) and reported by Pinheiro et al. (2011) in South America. Recently, based on satellite and ground-based observations in Uruguay and Southern Brazil, Bresciani et al. (2018) showed a decrease of ozone over these sites during October 2016 due to the influence of the AOH reaching mid-latitudes.

The air-mass transport in the stratosphere has been extensively analysed using the advected potential vorticity (APV) which is considered a suitable dynamical tracer in the stratosphere. The transport of polar air masses may take the form of
"filaments" or "tongue". These terms have been used to explain the transport of air from the edge of the polar vortex into middle latitudes by Waugh (1993) analysing potential vorticity maps, and previously by Randal et al. (1993), to explain the intrusion of tropical air into mid-latitudes. When the intrusion of air from the polar vortex reaches mid-latitudes and produces ozone decreases, it induces anomalies on the surface UV radiation. Bittencourt et al. (2018) also linked the occurrence of this event over South America to later changes in the tropospheric and stratospheric dynamic behaviour. Thus,
this parameter can be used to study the dynamics of the Antarctic polar vortex and as a tracer of poor-ozone air masses that are released from the AOH (Bittencourt et al., 2018; Kirchhoff et al., 1996, Pinheiro et al., 2011; Wolfram et al., 2012; Hauchecorne et al., 2002; Marchand et al., 2005; Bencherif et al., 2007).

In this paper we analyse an unusual event of rapid decrease and recovery of volume mixing ratio over Río Gallegos, Argentina, during November 2014 due to the release of a tongue of a poor-ozone air mass. This analysis was achieved by

means of ground and space-based instruments, focusing on the MWR ozone measurements. The high temporal resolution (one hour) of the MWR observations are analysed at different altitudes (27 km and 37 km) with the aim to determine the short-term variability of ozone mixing ratio and the moment when the polar vortex and its edge (as tongue or filamentary structure) with poor-ozone air masses pass over Río Gallegos and leave it at those altitudes, resulting in a local peak of ozone

mixing ratio for a very short period of time on November 2014. TOC measurements are also analysed by the ground-based instrument SAOZ installed in OAPA and by the satellite Ozone Monitoring Instrument (OMI). Finally, the APV field from the MIMOSA model was used to analyse the air-mass transport during the event. In addition, the MWR ozone mixing ratio retrieved in Río Gallegos is compared for the first time with ground-based measurements from the ozone DIAL/NDACC instrument and satellite measurements from the MLS on board the AURA/NASA.

This paper is organized as follows: section 2 describes the ground- and satellite-based ozone instrument, and the MIMOSA model used to calculate APV to determine the origin of air masses over Río Gallegos. In addition, this section describes the instrumental datasets used in this research and the methodology to analyse the correspondence of the MWR with respect to the ground-based DIAL instrument and the MLS ozone profile at the analysed altitudes. The results of the comparisons are detailed in Section 3. In section 4, the atypical ozone event occurred during November 2014 was analysed at 27 and 37 km

with a resolution of one hour determining the rapid variation of ozone mixing ratio over Río Gallegos. Finally, the solar Ultraviolet Index (UVI) at surface is also analysed during the event.

## 2 Materials and Methodology

### 2.1 Observations

Ground-based instruments used here are operated at OAPA, Río Gallegos, Argentina (51.5° S; 69.3°W), belonging to

CEILAP (hereafter OAPA). The geographical location of OAPA makes it a suitable site to study the sub-polar stratospheric ozone due to its closeness to Antarctica. Since 2005, a Differential Absorption Lidar (DIAL) has been operated at the OAPA with the aim to retrieve stratospheric ozone profiles (Wolfram, 2006; Salvador, 2011), which were joined to the Network for the Detection Composition Change (NDACC) in 2008 (http://www.ndsc.ncep.noaa.gov). In addition, a ground-based SAOZ spectrometer instrument (Pommereau and Goutail, 1988) to retrieve TOC was installed in the OAPA by 2008, which belongs

to LATMOS/CNRS. To contribute to ozone monitoring, the Solar Terrestrial Environment Laboratory, Nagoya University, Japan, installed the MWR in OAPA in 2011, which incremented the temporal resolution and increased the altitude range of the ozone profiles (Orte et al., 2011; Orte 2017). Due to the relationship between ozone amount and the solar UVB radiation at surface, this parameter is also measured in the OAPA with a ground-based solar radiometer YES UVB-1 (Yankee Environmental System, Inc.).

It is important to highlight that the MWR installed in the OAPA is one of few ground-based radiometers able to observe ozone in the southern hemisphere and the unique installed in subpolar region. In this hemisphere, other ozone radiometers can be found in the Antarctic region in Syowa station and in Halley stations (moved from the Troll station in 2013) (Isono et

al., 2014; Daae et al., 2014), and at mid-latitudes, in Lauder, New Zealand (McDermid et al., 1998). The MWR installed in the OAPA, allow to improve the understanding of the stratospheric and low-mesospheric dynamic using the ozone mixing ratio as a tracer and improving the validation of dynamical models at these latitudes.

Satellite instrument used here are other useful dataset to measure ozone on global scale and analyse the ozone layer behaviour. In this work, satellite OMI and MLS datasets are used to inter-compare with the ground-based MWR instrument.

### 2.1.1 Millimeter Wave Radiometer

The MWR is a fully automated instrument which belongs to Nagoya University. It retrieve ozone profiles ranging from ~25 and ~70 km vertical range with a temporal resolution of the order of one hour, allowing for the study of the short-term variability of this gas. The vertical resolution ranges from ~10 to ~14 km up to 48 km in high, increasing to 16 km above the middle mesosphere.

The MWR system is based on a superheterodyne receiver employing a superconductor-insulator-superconductor (SIS) mixer cooled at 4 K used to convert the ozone signal at ~110.83 GHz down to the intermediate frequency.

The MWR is basically composed of a rotating mirror, a quasi-optical mirror system, a superheterodyne receiver and a spectrometer. Figure 1 shows a scheme of the system installed at the OAPA. The rotating mirror looks toward four directions to acquire the signal from two different zenith angles, $S_{low}$ and $S_{high}$, and from two known reference blackbody loads to calibrate the signal from voltage to brightness temperature. $S_{high}$ comes from the zenith and it is re-directed to the rotating mirror by a fixed plane mirror, while $S_{low}$ comes from a zenith angle of between 12° and 38°. A dielectric plate is installed through the $S_{high}$ path to increment the continuum levels of the spectrum and then a servosystem is in charge to equalize both signals. A full description of the measurement technique can be found in Mizuno et al., 2002 and Parrish et al., 1988.

The calibration loads consist of two blackbodies at different temperatures, hot and cold. The hot blackbody load is achieved using a radio absorber at room temperature (~300 K), while the cold load is achieved by soaking a similar absorber in Liquid nitrogen (77 K) contained in a glass Dewar (vacuum bottle made of glass that is used especially for storing liquefied gases). The liquid nitrogen is obtained automatically using a compressor-refrigerator of environmental nitrogen. To reduce the standing waves (baseline), a pass length modulator (PLM) is inserted in the signal path which consists of a pair of roof-top mirrors (Mizuno et al., 2002).

The receiver is basically composed of a local oscillator (LO) and the SIS mixer (Ogawa et al., 1990). This system converts the input signal emitted by the atmospheric ozone molecules in their rotational transitions (~ 110,836 GHz) into the lower intermediate frequency (IF~6 GHz). The mixer operates in single side band (SSB) using a wave guide to filter the image band (Asayama et al., 2015) and it is cooled at 4K to reach the superconductive state. This operational temperature improves the signal to noise ratio to obtain a high temporal resolution (~1 hour). The temperature is achieved using a liquid helium closed loop cryogenic refrigerator (DAIKIN CG308, 3-stage GM-JT).

Then, the IF is amplified by a HEMT (High Electron Mobility Transistor) cooled to 15 K. The subsequent components (filters, amplifiers and attenuators) are designed to process the IF signal and fit it to the spectrometer requirements. The spectrometer is a digital FFT (DFS) Acquaris AC240 with 16384 channels and a bandwidth and spectral resolution of 1 GHz and 68 kHz, respectively. Finally, the observed spectrum in brightness temperature is obtained, assuming a linear behaviour among the sky signal ($S_{low}$ and $S_{high}$), the hot blackbody signal ($S_{hot}$) and the cold blackbody signal ($S_{cold}$) measurements by the following expression:

$$T_{oi} = \frac{T_{hot} - T_{cold}}{S_{hot} - S_{cold}} \left( S_{low} - S_{high} \right), \tag{1}$$

The method adopted here for the ozone profile retrieval is the optimal estimation method (OEM) described by Rodgers (Rodgers, 2000).

The forward model comparable to the measurement is calculated by means of the Atmospheric Radiative Transfer Simulator (ARTS). Detailed documentation can be found in http://www.radiativetransfer.org/docs /. The Qpack2 (Eriksson, 2005) is a package of Matlab routines used to setup the ARTS model and it has included the OEM calculation for general cases.

The input pressure and temperature for the forward model were obtained combining the NCEP reanalysis data up to ~30km with CIRA climatology above, interpolated for the MWR measurement time and site location. As input a priori ozone profile, we used monthly zonal daytime and night-time MLS O3 climatology between ~15 and 75 km, completed with zonal climatology below and above (McPeters et al., 2012). A full description of the inversion can be found in Orte, 2017.

### 2.1.2 DIAL (Differential Absorption Lidar)

The ozone DIAL used here was developed by the Centro de Investigación en Láseres y Aplicaciones (CEILAP) in collaboration with the Laboratoire Atmosphères, Milieux, Observations Spatiales (LATMOS; http://www.latmos.ipsl.fr), between 2003 and 2005 (Wolfram, 2006; Salvador, 2011). The ozone profile retrieval algorithms were provided by the Observatory of Haute-Provence (OHP) (Godin, 1987; Godin-Beekmann et al., 2003; Pazmiño, 2003), which were adapted for the DIAL system installed in the OAPA, Río Gallegos (51.6° S; 69.3 ° W) by mid-2005. Since the installation, instrumental and algorithm improvements have been carried out (Salvador, 2011).

DIAL is an active and self-calibrated remote sensing technique similar to radar but using pulses of laser radiation in the ultraviolet range. This instrument requires the emission of two lasers directed to the atmosphere in two different wavelengths: 308 nm ($\lambda_{on}$) and 355 nm ($\lambda_{off}$). A ClXe excimer laser is responsible for emitting the laser beam at the wavelength $\lambda_{on}$, which is absorbed by the atmospheric ozone molecules, while $\lambda_{off}$ is the reference wavelength produced by the third harmonic of an Nd-YAG laser. The interaction of this laser's radiation with the atmospheric molecules causes scattering following a known spatial distribution, and the photons backscattered in the direction of the instrument are collected by four Newtonian telescopes with a diameter of 50 cm each, which have an aluminized reflective parabolic surface. These photons are reflected and focused on four optical fibers, each located in the focus of each parabolic mirror.

The photons are conducted to a mechanical chopper positioned before the spectrometer to filter the backscattered lidar signal from the bottom of the atmosphere. Finally, a spectrometer is used to separate the backscattered signal at different wavelengths. These signals are then integrated in time (~3 hours) and processed by the retrieval algorithm to obtain ozone profiles. The DIAL instrument covers an altitude range from ~ 15 to ~ 40 km under optimal operating conditions with a vertical resolution between 0.5 and 5 km, depending on the altitude, and it can only operate during clear sky nights. The typical uncertainty associated to this instrument varies between 3 to 15 % from 14 km to 35 km (Wolfram et al., 2012).

### 2.1.3 AURA satellites: MLS and OMI

The MLS (Microwave Limb Sounder) was launched on July 15, 2004 on board the AURA satellite and it began to operate on August 13, 2004. Since then, this instrument has been able to observe the thermal emission of the atmosphere in the range of submillimeter and millimeter waves. The Earth's limb viewing allows the MLS to achieve a higher altitude resolution (~3 km in the stratosphere and ~5 km in the mesosphere) compared to MWR. A full description of the MLS instrument can be found in Waters et al. (2006). The MLS ozone profiles data versions 3.3 and 3.4 (Livesey et al., 2013) were used (http://mls.jpl.nasa.gov).

The OMI, on board the Aura satellite, started TOC measurements in 2004, with the aim to continue the TOMS satellite record. It was launched in July 2004 in the framework of the Earth Observing System (EOS) project. In addition to ozone, OMI instrument retrieves atmospheric components such as total contents of $NO_2$, $SO_2$, aerosols, among others. This instrument measures the reflected and backscattered solar radiation by an UV-VIS spectrometer with a spectral resolution ranging from ~0.45 to ~0.63nm in nadir view and provides nearly global coverage in one day with a spatial resolution ranging from 13 to 24 km (Levelt et al., 2006). In this work, we used the TOC overpass product from OMI (OMDOAO3). The dataset can be downloaded from https://avdc.gsfc.nasa.gov/index.php?site=2045907950.

### 2.1.4 SAOZ (Systeme d'Analyse par Observation Zenithale)

The ground-based SAOZ UV-VIS (300 – 650 nm) spectrometer instrument (Pommereau and Goutail, 1988) used in this work was installed in the OAPA observatory in March 11, 2008 and it belongs to LATMOS/CNRS. SAOZ measures the sunlight scattered from the zenith sky. Differential Optical Absorption Spectroscopy (DOAS) method is applied to retrieve total ozone and nitrous dioxide columns twice a day at high solar zenith angles between 86º and 91º at sunrise and sunset. TOC-measurements are performed in the Chappuis visible band (450-550 nm) where ozone cross section are little dependant on temperature. The spectral resolution of the SAOZ installed at Río Gallegos is 0.9 nm. SAOZ retrieval follows UV-Vis NDACC Working Group recommendations: spectral window analysis, absorption cross sections and daily air mass factor to convert measured slant column densities (SCD) in vertical column densities (VCD). In the case of ozone, look-up tables (LuT) of air mass factors are used (Hendrick et al., 2011). The LuT were obtained from the UVSPEC/DISORT radiative transfer model using the TOMS V8 ozone and temperature profiles climatology. This SAOZ joined the NDACC network in

2009 and the dataset can be downloaded from SAOZ webpage (http://saoz.obs.uvsq.fr/SAOZ-RT.html) and NDACC webpage (ftp://ftp.cpc.ncep.noaa.gov/ndacc/station/gallegos/ames/uvvis/).

### 2.1.5 Solar Radiometer YES UVB-1

The ground-based radiometer YES UVB-1 (Yankee Environmental System, Inc.) installed in OAPA is used to measure the
UVB irradiance at surface, and the UVI is retrieved. It is connected to a data logger, which is configured to acquire one measurement per minute. Due to the UVI is strongly affected by the ozone amount in the atmosphere, the time evolution of the daily maximum UVI is analysed during the period of the case study. We decided to present the daily maximum UVI instead of the UVI at solar noon due to the fact that most of the analysed days during the low ozone event were partially cloudy and the maximum UVI were observed near the solar noon. Thus, the daily maximum UVI is more representative in
terms of the low ozone amount impact over the UVI at surface.

### 2.2 MIMOSA Model

The Modélisation Isentrope du transport Méso-échelle de l'Ozone Stratosphérique par Advection (MIMOSA) high-resolution advection model was used here to determine the origin of air masses similar to an isentropic Lagrangian trajectory model. The MIMOSA dynamical model is specifically used to describe filamentary structure through the APV on isentropic levels
(Hauchecorne, et al., 2002; Godin et al., 2002). The advection is driven by ECMWF meteorological analyses at a resolution of 0.5°x0.5°. It is possible to run the model continuously and follow the evolution of PV filaments for several months. The accuracy of the model has been evaluated by Hauchecorne et al. (2002) and validated against airborne lidar ozone measurements using a correlation between PV and ozone (Heese et al., 2001; Godin et al., 2002; Jumelet et al., 2009). The ability of the MIMOSA model to determine the origin of air masses influencing a given site has been highlighted in several
studies (Hauchecorne et al., 2002; Bencherif et al., 2003; Jumelet et al., 2009; Bègue et al., 2017). Moreover, the MIMOSA model is frequently used to detect the origin of air masses inducing laminae on ozone profiles (Hauchecorne et al., 2002; Godin et al., 2002; Portafaix et al., 2003). A full description of this model can be found in Hauchecorne et al., 2002.

### 2.3 Methodological considerations

Fifteen months of MWR, MLS and DIAL ozone measurements at different altitude over OAPA were analysed, from October
2014 to December 2015. Figure 2 shows the time series of ozone mixing ratio observed by the MWR (blue) and the MLS (red) for altitudes of 27, 37, and 65 km. The first two levels are established in such a way that they are representative of the amount and variability of ozone within the stratosphere, in the dynamic range of the instrument. In addition, around the 37 km occur the maximum of the average ozone mixing ratio for Río Gallegos. The level of 65 km was included to observe the sensitivity of the MWR in the mesosphere within the diurnal cycle. We observe a marked difference of ozone mixing ratio
between day and night measurements due to the ozone photochemistry around this altitude (Allen et al., 1984, Nagahama et

al., 1999). In general, we can observe that the behaviour of the MWR and MLS measurements for all analysed altitudes is similar.

The MWR has a temporal resolution of ~1 hour, while the MLS presents measurements close to the OAPA with an approximate frequency of one measurement every two days at 19:00 UTC approximately, and two monthly measurements at 5:00 UTC approximately. This frequency is conditioned by the orbit of the AURA satellite. In order to obtain a significant number of profiles to make the comparison, the MLS observations were selected within a box of ±0.2 in latitude and ±5° in longitude from the OAPA location, considering that both instruments were observing the same air mass. Figure 3 shows the position of the MLS measurements selected for the inter-comparison (yellow crosses) and the location of the OAPA (blue dot), where the MWR is located. Numbers below crosses indicates the number of each group of MLS measurements in each location. The time differences between MWR and MLS measurement inter-comparison pairs were less than 30 minutes. Given that MLS measurement are not collocated with the MWR, some differences between instrument could be due to the distance between instruments, mainly during spring when the AOH may influence and produce large difference of ozone mixing ratio in short distances.

While the MWR and the MLS operate automatically, the DIAL requires manual operation in clear sky nights. DIAL monitoring in 2014 and 2015 was intensive during spring (October, November, December) and it was possible to obtain a few measurements if we compare with the high quantity of measurements provided by the MWR and the MLS. The DIAL dataset is not shown in figure 2, but in figure 6 in the next section. For the MWR – DIAL intercomparison pairs we take the MWR measurements in the middle of the ~3 hour integrated interval of the DIAL measurement.

The ozone measurements obtained by the DIAL are in molecules/m3, while the MWR and MLS measurements are in volume mixing ratio. Thus, temperature and pressure from NCEP reanalysis data are then used to convert from molecules/m$^3$ to volume mixing ratio.

Due to the fact that MLS and DIAL have a better vertical resolution than the MWR, the MLS and DIAL profiles were degraded to the MWR resolution taking into account the averaging kernel functions A, which represent the response of the retrieved ozone profile to the "true" one. The following expression was used to degrade the vertical resolution (Palm et al., 2010):

$$x_{LR} = x_a + A(x_{HR} - x_a) \tag{2}$$

where, $x_{HR}$ is the MLS or DIAL ozone profile (depending on which instrument is intercompared with the MWR) with the original vertical resolution. $x_a$ and A are the a priori ozone profile and the averaging kernel function used in the MWR inversion, and $x_{LR}$ are the MLS or DIAL ozone profiles degraded to the MWR vertical resolution. Since the ozone DIAL profile is limited in altitude range respect the millimter wave radiometer, it is completed below and above of the measurement with MLS climatologies, interpolated to the OAPA location.

The main source of millimeter wave opacity that impacts radiation coming from ozone molecules in the stratosphere and the mesosphere is water vapour, mainly contained in the troposphere. The opacity is retrieved by the MWR during the

measurement cycle (Orte, 2017). Only measurements taken when atmospheric opacity was less than 0.29 were considered. This criterion was defined taking into account the mean value and the variability of the opacity for Río Gallegos, measured by the MWR ($\mu_\tau = 0.225; \sigma_\tau = 0.041$) for the analysed period and studying the correlation between MLS and MWR. We noted that the correlation between MLS and MWR at different altitudes increases when opacity decreases (not presented

here).

For evaluating the correspondence between instruments, the mean bias error (MBE) was calculated between the MWR and the DIAL and MLS measurements ($x_{LR_A}$) for the considered altitudes (27 km, 37 km and 65 km):

$$MBE = 100 \; x \; \frac{\overline{MWR_A} - \overline{x_{LR_A}}}{\overline{x_{LR_A}}}$$    (3)

where $\overline{MWR_A}$ is the average of the MWR ozone mixing ratio at each altitude and $\overline{x_{LR_A}}$ is the average of the DIAL or MLS

measurements at the same altitude.

Finally, a linear regression analysis between each datasets pair at each altitude was performed and the correlation coefficient R was analysed.

## 3. Results

### 3.1 Inter-comparison of MWR with DIAL and MLS observations

With the aim of analysing the correspondence of the MWR with independent instruments, inter-comparisons of ozone mixing ratio respect to the ground-based DIAL instrument and satellite-based MLS were carried out at 27, 37 and 65 km.

Figure 4 shows the number of time overlap measurements inter-compared between MWR and MLS (blue bars), and MWR and DIAL (light blue bars) during the period described in Figure 2. A total number of 84 MWR-MLS and 30 MWR-DIAL measurements pairs were inter-compared during the analysed period. The number of MWR-DIAL pairs is larger during

spring (Sep-Oct-Nov). This is because the DIAL measurement campaigns become more intense in those months when the AOH approaches and overpasses the southern South America (Wolfram et al., 2012). Between December 2014 and July 2015, there were no DIAL measurements. On the other hand, we observe that the number of MWR-MLS inter-compared pairs during spring and summer amounting to 59, was larger than that during autumn and winter with 25 pairs. February, March and July do not present inter-compared measurements because few MWR observations were retrieved and did not

match MLS observations according to the spatial and temporal overlap criteria defined in the methodology section.

### 3.1.1 MWR – MLS comparison

Figure 5 (left) shows the time series of the MLS and MWR ozone mixing ratio for 27, 37 and 65 km for measurements when the opacity was less than 0.29 (See Section 2.3). The behaviour of both data series is similar for all altitudes considered. Figure 5 (right) presents the scatter plot between both instruments at different altitudes and the linear regression together

with the correlation coefficient (R). Table 1 summarizes the results of the comparisons between datasets.

We compared N=84 MWR – MLS measurements pairs, taking into account the spatial selection criteria according to the location of the MLS measurement with respect to the location of the MWR (LatOAPA ± 0.2°; LongOAPA ± 5°).

The linear regression analysis at 27 km presents a slope of 1.01 and an intercept value of 0.25. The correlation coefficient (R) of 0.65 reflects considerable correlation for both datasets. The MBE was calculated to analyse the bias between ground-based and satellite data. We obtained a value of +5% indicating an MWR overestimation with respect to the MLS.

Unlike the average ozone mixing ratio at 27 km, the MBE at 37 km reflected an underestimation of ozone mixing ratio of -11% compared with MLS. Fiorucci et al. (2013) also presented differences ranging between -8% and -18 % in the 17–50 km vertical range, reaching ~-18% at 37 km. The regression analysis presents a slope of 0.96 and an intercept of 0.44. Similarly, the correlation coefficient at this altitude was calculated (R = 0.63) to evaluate the correlation between MWR and MLS at this altitude.

The best correspondence was found at 65 km. The linear regression presents a slope near to the unity (0.95) with an intercept close to zero (-0.02 ppm). The correlation between measurements was also close to unity (R=0.88), which reflects very good agreement. Finally, a MBE value of -7% shows an underestimation of the average MWR measurements in comparison to MLS.

The difference between measurements can be attributed to the typical uncertainties of each instrument, although another source of difference could be explained by the non-collocated measurements inter-compared. This point is discussed in section 5.

### 3.1.2 MWR – DIAL comparison

Figure 6 (left) shows the ozone mixing ratio measured by the MWR and the DIAL for 27 km at the same time, and a comparison between both instruments by mean of a scatter plot (right).

The slope and intercept in the linear regression were 0.93 and 0.36 ppm (~6% of the observed average mixing ratio), respectively, with an acceptable correlation coefficient (R=0.73) (Table 1). This reflects a good agreement between both ground-based instruments at 27 km. Unlike the MWR – MLS inter-comparison at 27 km (R = 0.65), MWR and DIAL instruments are installed at the same place which might explain, in part, the better correlation. The observed discrepancy can be attributed to instrumental uncertainties.

### 3.2 Short term ozone variability

To study the short term ozone mixing ratio variability related to the AOH influence over Río Gallegos, an extreme event of rapid variation occurred between November 15 and 20, 2014, was analysed using the MWR measurements at 27 and 37 km. A 3 hours running mean was applied.

Zonal ozone mixing ratio climatologies were calculated from the MLS ozone profiles measurements from 2004 to 2016 at both altitudes, interpolated to the latitude of Río Gallegos to analyse differences between measurements and mean values presented at those latitudes.

Finally, the APV was calculated using the MIMOSA model for these altitudes to interpret the ozone measurements.

### 3.2.1 Description of the case study

After analysing the correspondence of the MWR measurements with independent instruments, here we analyse a short-term ozone variation for an atypical case study of the influence of the AOH over Río Gallegos during November 2014.

Figure 7a (bottom) shows the time evolution of the MWR ozone mixing ratio at 27 (blue line) and 37 km (red line) by November 2014 with their respective zonal mean value (white line) and one standard deviation (SD) (light blue and light red, respectively). The statistical quantifiers were calculated from the MLS profile dataset, interpolated to the latitude of Río Gallegos from 2004 to 2016. Both altitudes present similar behaviours. We observe a rapid ozone decrease at both altitudes from November 11 at 19:30 local time (LT) to November 15. The minimum value at 27 km is reached at 6:30 LT, while at

37 km it occurs at 5.30 LT, and both minimums are far than two SD from the mean value (around less than 25 and 20% respect the climatology, respectively). This decrease is related to the influence of the AOH over Río Gallegos, followed by a rapid increase reaching a pick on November 17 at 14:30 at both altitudes. At 27 km, the maximum (~6.1 ppm) reaches values above the mean, while at 37 km (~6.6 ppm) it does not reach the mean. After that, the ozone mixing ratio presents a new local valley reaching the minimum on November 19 at 01:30 LT.

Figure 7a (top) also shows the time evolution of TOC from the OMI and the SAOZ installed in the OAPA. The mean value and the SD are depicted by the white line and the shadow area, respectively. We can observe the difference in the frequency of measurements (lower time resolution) with respect to the MWR observations. The general behaviour of both measurements follows the behaviour of the MWR at 27 and 37 km and it shows the influence of the ozone hole on the TOC with a valley from ~November 11 to ~November 22, where the TOC reached unusual values of ~230DU by November 14

(~30% below from climatology) and it is below one SD in the whole mentioned period. The OMI measurements did not present the local atypical maximum described above because its time resolution did not allow observing it. This atypical event is presented in the SAOZ measurement between November 16 at 21:00 and November 17 at 21:00, although the TOC was below one SD from the mean value.

  Figure 7b (blue dots) presents the time series of the daily maximum Ultraviolet Index (UVI) (near the solar noon) during the

low ozone event described before measured with a radiometer YES UVB-1 (Yankee Environmental System, Inc.) installed in the OAPA. White line and shadow area are the climatological UVI at noon calculated from the overpass UVI of satellite OMI instrument between 2004 and 2017 for Río Gallegos. As expected, we observe that the UVI presents an opposite behaviour respect the SAOZ and OMI TOC measurements. It is important to note that on November 17 the maximum UVI presents a local minimum value close to the climatology. This local minimum on the maximum UVI could be associated

with the short-term ozone recovery observed in the MWR measurement mentioned above (local peak on November 17).

### 3.2.2 Dynamical context. AOH influence

In order to determine/confirm the polar vortex influence over Río Gallegos and explain the behaviour of the MWR measurement at 27 and 37 km peaking on November 17, the Advected Potential Vorticities (APV) from the MIMOSA model were analysed at 675 K (~27km) and 950K (~37km) for the same period (Figures 8 and 9). Figure 8 show the APV for the Southern Hemisphere to describe the state of the polar vortex at both altitudes and the recovery during a short period from November 11st to 18th, while figure 9 presents the APV over Río Gallegos at both isentropic levels.

On November 11, the polar vortex is out of the continent for both isentropic levels. From November 13rd to December 16th, we observed low values of APV over Río Gallegos, which is correlated with the decrease in ozone amount for the MWR at 27 and 37 km (Figure 7), and with the decrease in the TOC from OMI and SAOZ, due to polar air masses. On the 17th, the APV map shows the formation of a yellow tongue at 675 K reflecting the lower values of APV between ~-39° and ~-43° of latitude over the continent. In the APV map at 950K, some blue filaments with low APV values can be observed at similar latitudes. We observe that Río Gallegos is out of this tongue and filaments on November 17, and air masses from outside the polar vortex were passing over Río Gallegos. On November 18th, poor ozone air masses reach Río Gallegos again, and the increase of ozone mixing ration at both altitudes is observed in Figure 7.

The APV over Río Gallegos (figure 9) at 675K (~27km) and 950K (~37km) presents a similar behaviour of the MWR measurements at 27 and 37 km. At both altitudes, the APV decreased from November 12nd to 14th, with an increase around 17th, followed by a new decrease on November 18th. For 950 K, the increase before November 17th is smoother than at 675 K, which gives account that the polar vortex at 37 km reached Río Gallegos earlier than at 27 km.

Both analyses in figures 8 and 9 confirm that the polar vortex was retired from Río Gallegos for a short period around November 17th, which explains the local maximum in the ozone mixing ratio detected by the MWR for both altitudes in Figure 7. Thus, the high time resolution of the MWR measurements enables to observe the short term ozone variation and determine the influence of the AOH over Río Gallegos at each altitude, with a time resolution of one hour, when the atmospheric conditions allow taking measurements.

### 4. Discussion

It is well known that the southern part of South America is affected by the frequent abrupt intrusions of the AOH during the spring (Wolfram et al., 2012; Kirchhoff et al., 1997; WMO, 2013; WMO, 2012; WMO, 2011b). As a consequence of this phenomenon, the ozone amount in the middle atmosphere suffers sudden variations in short time periods of the order of hours. In this paper we presented a case study of short term ozone mixing ratio variability at different isentropic levels over Río Gallegos, Argentina, during November 2014, as consequence of the Polar vortex influence over this region. The study could be conducted thanks to the high time resolution of the MWR instrument used. The influence of the polar vortex during the analysed period was confirmed in the APV from the MIMOSA model at two isentropic levels (675K and 950K). We observed filaments of air-mass from the polar vortex at both potential temperature levels passing over Río Gallegos. Similar

cases of filaments travelling toward mid latitudes in the South Hemisphere have been reported analysing the APV (Waugh, 1993) without the possibility to report the stratospheric ozone amount with the time resolution reported here.

Other satellite or ground-based instruments that monitor the vertical ozone amount, such as MLS or DIAL, have lower time resolution and they are not able to observe the short-term ozone variability. This fact shows the capability of the MWR and the needed to retrieve the ozone mixing ratio at high time resolution to analyse the short-term variability in these regions directly affected by the passage of the polar vortex at different altitudes.

In addition to the short-term ozone recovery, during the analysed period was observed reductions as consequence of the ozone hole influence. The ground-based SAOZ and satellite OMI instruments reflected maximum reduction of around 30% in TOC. Similar reduction has been found in Wolfram et al. (2012) during November 2009, while Kirchhoff et al. (1997) had reported maximum reduction of around 60% by 1992-1994 at similar latitudes (Punta Arenas, Chile) respect the monthly mean values. If we analyse the ozone reduction in altitude, we observed maximum decreases of 20% and 25% respect the climatology value at 27 km and 37 km, respectively. DIAL measurements of ozone profiles carried out in the OAPA have shown maximum differences of around 50% in September-November (WMO, 2013; WMO, 2012; WMO, 2011b). These results highlight the importance of measurements at sub-polar regions.

The time evolution of the daily maximum UVI was also analysed during the study period. As expected, we find an opposite behavior respect the total ozone column, which is in agreement with other results reported (Casiccia et al., 2008; Wolfram et al., 2012). It is observed a local minimum when the measurements of the ozone amount retrieved by the MWR presents a maximum at both altitudes.

In sections 3.1, we evaluated the correspondence between the MWR with respect to the MLS at 27 and 37 km and with respect the ground-based DIAL at 27 km. The MWR-MLS inter-comparison at 27 km reveals a MBE of 5%, which is consistent with the value, obtained Ohyama et al. (2016). Boyd et al. (2007) also carried out similar inter-comparisons between MLS and two MWR installed in Mauna Loa, Hawaii and Lauder, New Zealand. The differences reported for Lauder range from +7% to 10% between ~20 to ~28km, while for Mauna Loa differences are around ~3% (Figure 1, Boyd et al. (2007)). On the other hand, Fiorucci et al. (2013) reported a difference of 10% at 26 km of altitude. Thus, the comparisons carried out between MWR and MLS reveal good agreement for the considered altitudes, consistent with the results of other authors.

Similarly, we analysed the MWR-DIAL comparison at 27 km and we can observe that the correlation coefficient (R = 0.73) and the MBE (1%) are consistent with those obtained by a similar inter-comparison carried out by other authors. Nagahama et al. (1999) obtained a correlation coefficient of 0.77 and a MBE=1%, although that analysis was realized at 38 km. Studer's et al. results reflect a 1.43% of difference between MWR and DIAL comparison.

When we compare the MWR with the MLS, it is considered that both instruments are measuring the same air masses, although the location of the satellite measurements differs from the location of the MWR measurements, which can introduce a difference in the ozone mixing ratio measured. These differences could be accentuated during the austral spring, when the AOH occurs, since ozone mixing ratio values can vary considerably over short distances.

At 27 km, we can observe that the correlation between MWR and DIAL (R = 0.73) is better than that between MWR and MLS (R = 0.65) at the same altitude. In addition, the slope for the linear regression analysis reflects a 1% relative difference between MWR and DIAL, while MWR presents a positive bias of 5% with respect to MLS.

One reason why the correspondence between the MWR and the DIAL is greater with respect to the MLS may be that the two instruments installed on the ground (MWR and DIAL) are monitoring the same air mass, while the distance with the location of the MLS observations could be introducing differences in the comparison. Figure 3 shows the position of the 84 MLS measurements analysed (yellow crosses) with respect to the location of the OAPA, where the MWR is located. Numbers below crosses indicates the number of each group of MLS measurements in each location. The maximum distance between measurements reaches ~341 km while the minimum distance is ~23 km with an average distance of 207 km. Only 22% of MLS observations are at a distance less than 100 km from the MWR, while more than 50% of the inter-compared observations are farther than 200 km. Therefore, the distance between the considered location of the MLS measurements and the location of the MWR could explain partly the difference between the ozone mixing ratios retrieved from these two instruments. Comparisons between DIAL and MLS were realized by Sugita et al. (2017) for an unusual case of persistence of the AOH over Río Gallegos occurred during November 2009, who also attributed part of the differences to the non-co-location of the measurements. Future studies analysing longer datasets will be interesting to determine the influence of the distance between measurements on the ozone mixing ratio differences between instruments in this region.

It is important to note that the MWR and DIAL instruments retrieve ozone in different fundamental units. While the MWR provides the ozone mixing ratio, the DIAL provides the ozone number density as a function of altitude. The DIAL unit was converted to the MWR unit for the inter-comparison using the temperature and pressure retrieved from the DIAL. Thus, uncertainties in these parameters could be adding uncertainties in the ozone amount in ppm from the DIAL.

## 5. Conclusion

We have presented ozone mixing ratio measurements at 27, 37 and 65 km with a temporal resolution of ~1 hour from a Millimeter Wave Radiometer installed at Río Gallegos, Argentina, from October 2014 to December 2015.

The MWR ozone mixing ratio retrieved was compared for the first time with ground-based measurements from the ozone DIAL instrument and satellite measurements from the MLS on board the AURA in defined overlap altitudes. The comparison revelled good correspondence between independent instruments. The comparison with MLS measurements presents a positive bias of 5% at 27 km and a negative bias of -11% and -7% at 37 and 65 km, respectively. The correlation between measurements at those altitudes was 0.65, 0.63 and 0.88 at 27, 37 and 65 km, respectively. The comparison with the DIAL data at 27 km reflected a good correspondence with a negative bias of -1% with a correlation coefficient of 0.73.

We observed better correspondence between MWR and DIAL at 27 with respect the MWR-MLS. One reason of this better correspondence may be that the two instruments installed on ground (MWR and DIAL) are monitoring the same air mass, while the distance with the location of the MLS observations could be introducing differences in the comparison.

Moreover, this work highlights the capability of the MWR installed in Río Gallegos for the determination of short-term variations of the ozone mixing ratio at different altitudes in this strategic location at the edge of the AOH, making it possible to detect the influence of this phenomenon as we showed in the atypical study case held on November, 2014. The rapid variation of ozone at 27 and 37 km was analysed in correspondence with the perturbation of the APV derived from the MIMOSA model which explain the volume mixing ratio peak due to the retired of the polar vortex for a short time on November 17. The time evolution of the daily maximum UVI measurements during the analysed period, clearly reflect the anti-correlation with the TOC.

The MWR installed in the OAPA cover the lack of ground-based radiometer observation of ozone between Antarctic latitudes and mid-latitudes, allowing to improve the understanding of the stratospheric and low-mesospheric dynamic using the ozone mixing ratio as a tracer and improving the validation of dynamical models. It is expected to join the MWR to the NDACC Network in future.

## Acknowledgement

This work was supported by the French-Argentine ECOS-Sud  A16U01 project. The authors would like to thank to JICA and JST for the financial support of SAVER-Net network and OAPA facilities in South of Argentina. Also, thanks a lot to the referees for their valuable comments which helped to improve the manuscript.

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

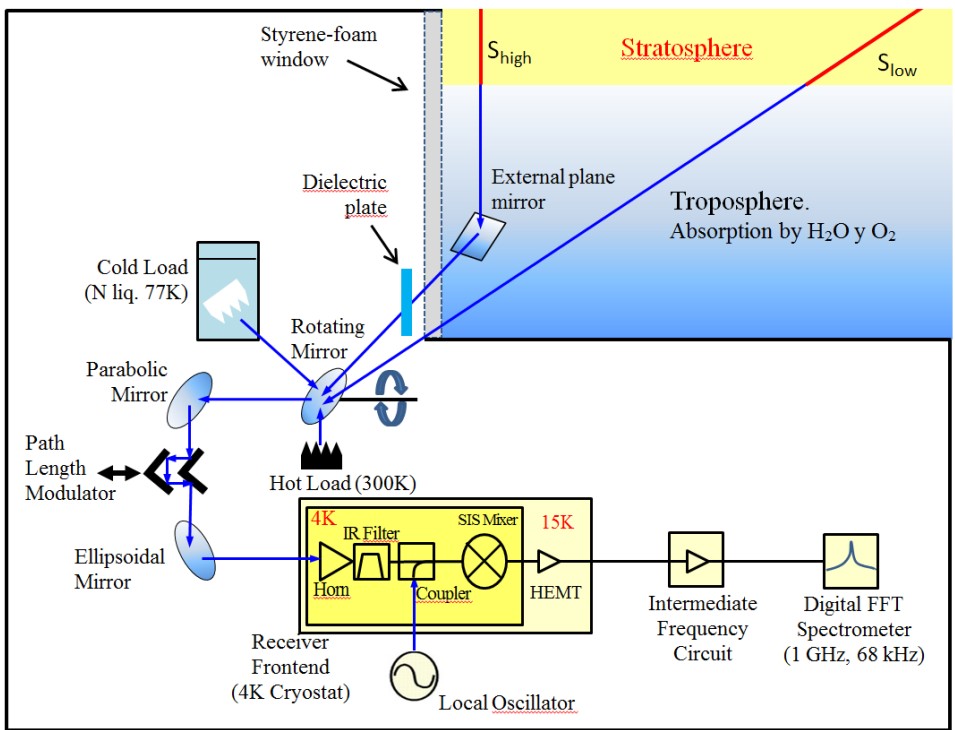

**Figure 1 Block diagram of the MWR at the OAPA, Río Gallegos (51.6°S, 69.3°W).**

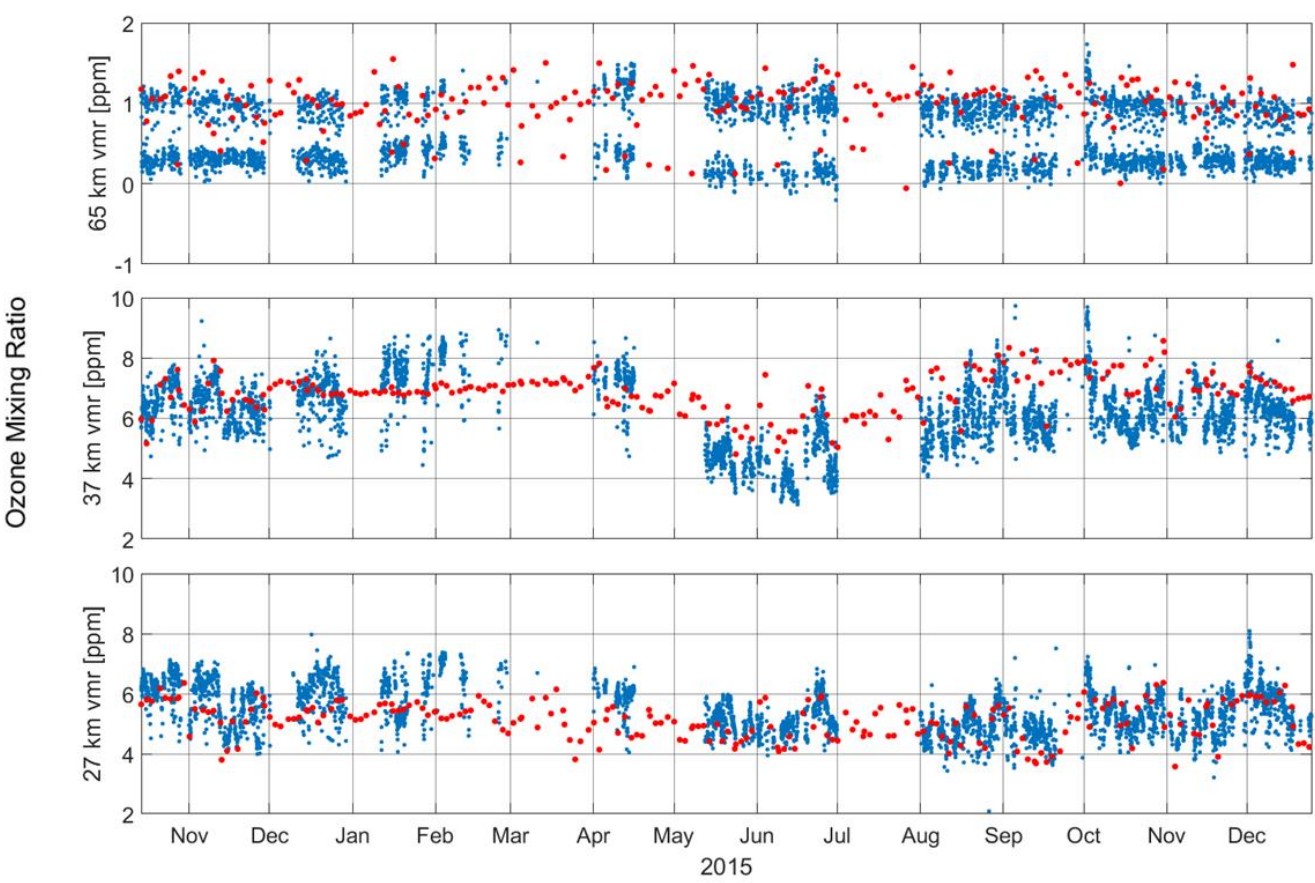

**Figure 2** Time series of MLS (red dots) and MWR (blue dots) ozone mixing ratio for three altitudes: 27, 37 and 65 km between October 2014 and December 2015 (the MWR was inoperative during March and July 2015 due to technical problems).

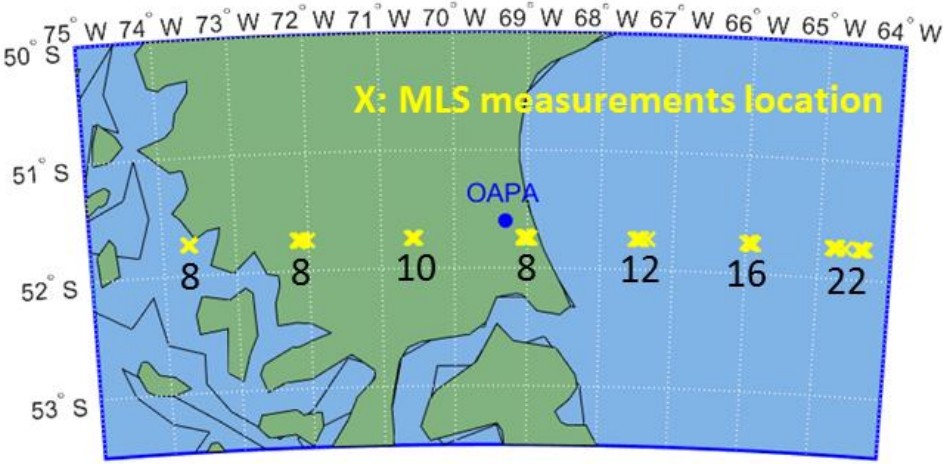

**Figure 3 MWR location (blue dot, OAPA) and MLS measurements location (yellow crosses) used in the inter-comparison. The numbers below crosses indicates the quantity of MLS measurements.**

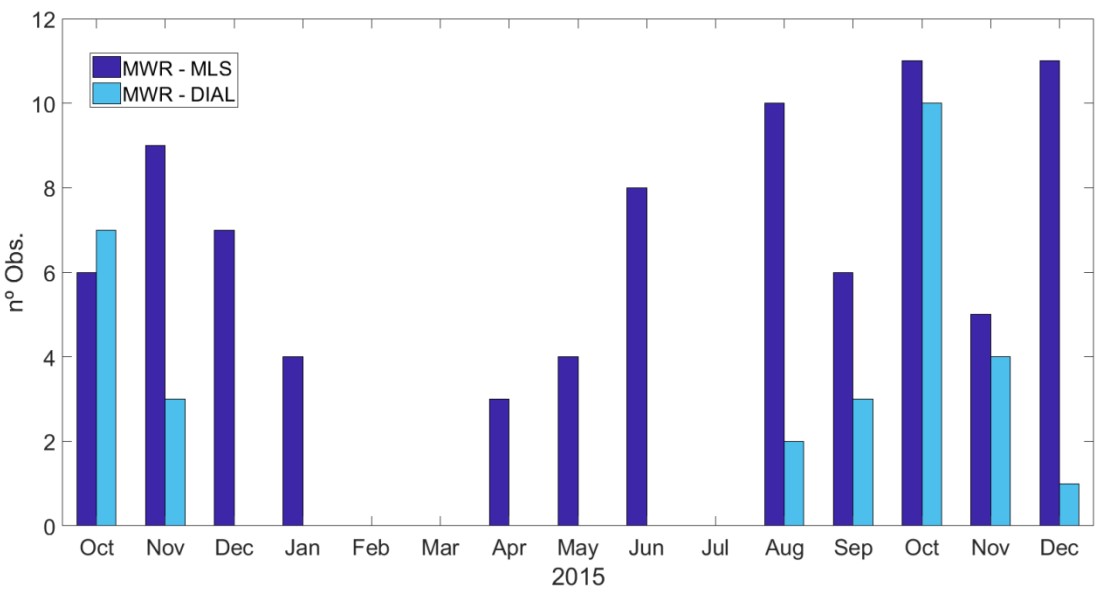

**Figure 4 Number of inter-compared measurement pairs for each month. Blue bars represent the number of MWR-MLS pairs while light blue bars are the number of MWR-DIAL pairs.**

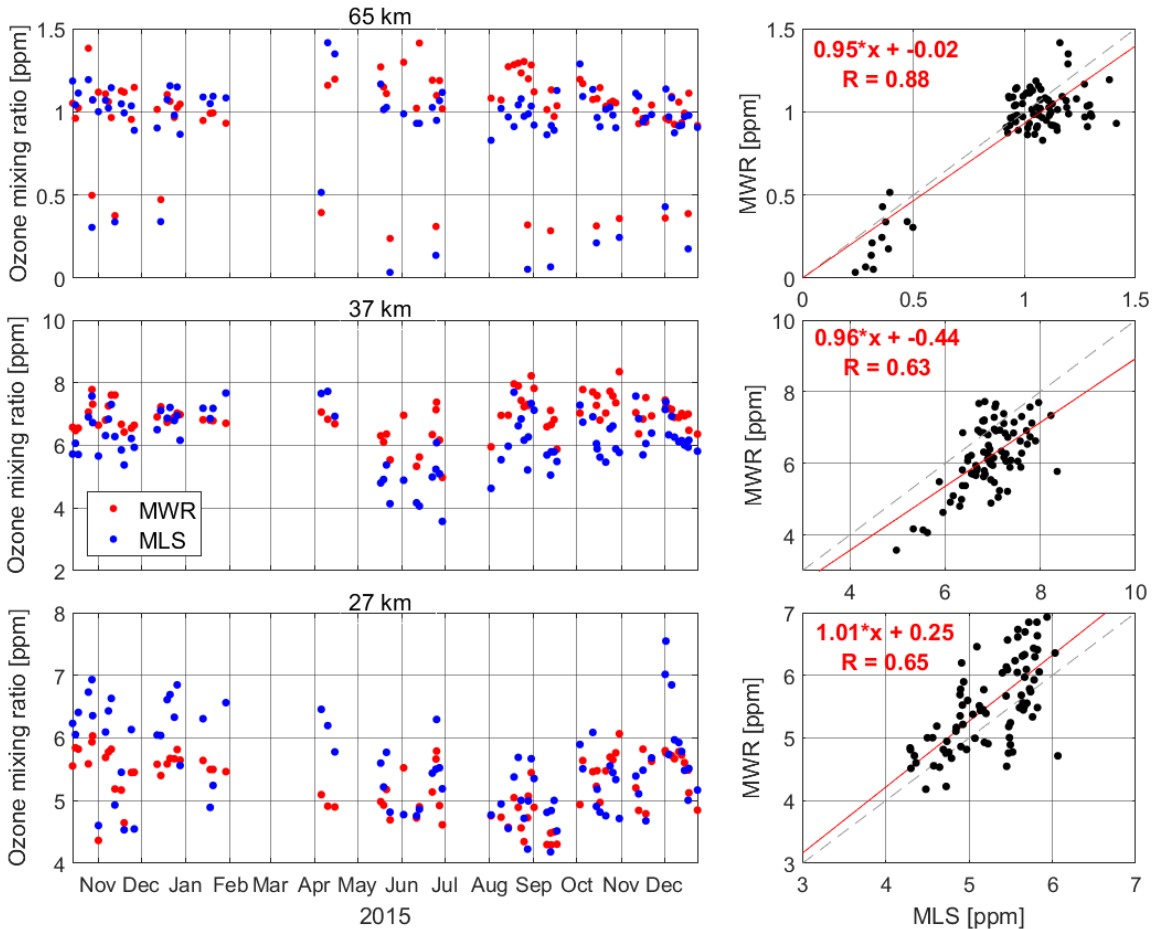

Figure 5 (left) MLS (red dots) and MWR (blue dots) ozone mixing ratio at the same time and within a box of ±0.2 in latitude and ±5º in longitude from the OAPA location for three altitudes: 27, 37, and 65 km. The analysed period covers from October 2014 to December 2015, with a total of 84 overlap measurements; (right) Scatter plot between MLS and MWR measurements for the three altitudes analysed.

| | Alt. | N | Slope | Intercept [vmr(ppm)] | R | MBE |
|---|---|---|---|---|---|---|
| MWR-MLS | 27 km | 84 | 1.01 | 0.24 | 0.65 | +5% |
| | 37 km | 84 | 0.96 | -0.43 | 0.63 | -11% |
| | 65 km | 84 | 0.95 | 0.02 | 0.88 | -7% |
| MWR-DIAL | 27 km | 30 | 0.93 | 0.36 | 0.73 | -1% |

**Table 1 Statistical parameters of the MWR respect the MLS and DIAL measurements intercomparison. N: number of intercomparison pairs; R: correlation coefficient; MBE: Mean Bias Error.**

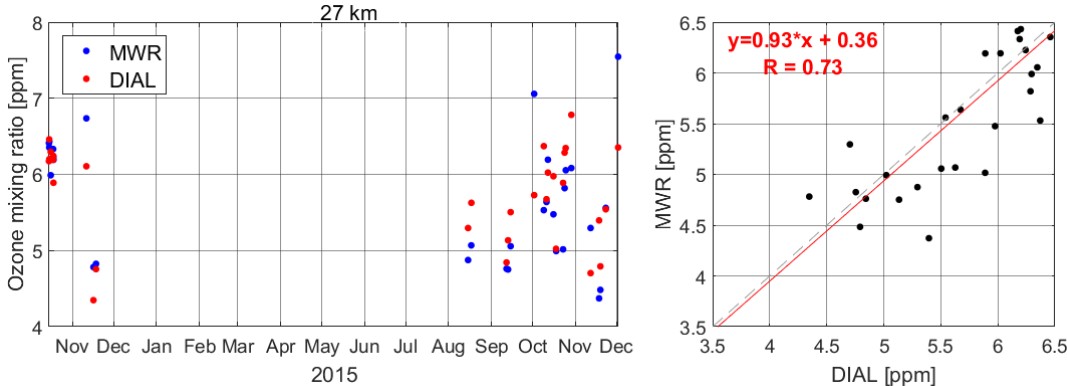

**Figure 6 (left) DIAL (red dots) and MWR (blue dots) ozone mixing ratio at the same time for 27 km between October 2014 and December 2015; (right) Scatter plot between DIAL and MWR measurement.**

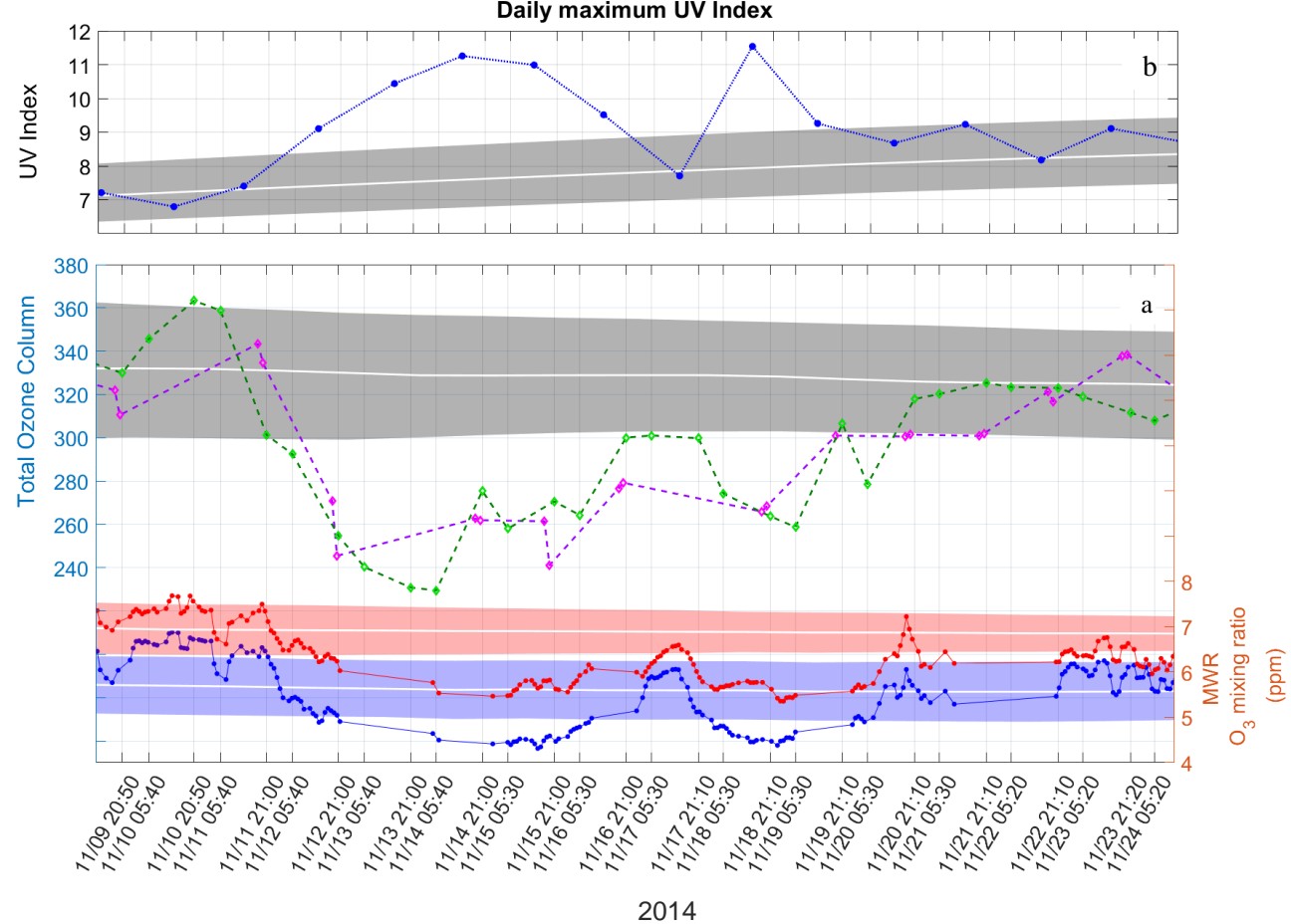

**Figure 7. Atypical event of Antarctic ozone hole influence over Río Gallegos. (a) (Bottom) Time evolution of the MWR ozone mixing ratio at 27 km (red line) and 37 km (blue lines). Light red and light blue areas represent the ozone mixing ratio zonal climatology at both altitudes calculated using MLS database (2004 - 2016). (Top) Time evolution of total ozone column measured with the ground-based SAOZ instruments (green dots) and OMI (purple dots) in Dobson Units. White line and grey area represent the climatology and one SD calculated using the OMI data-base (2004 - 2017). (b) Time Evolution of the Daily maximum Ultraviolet Index measured with the ground-based solar radiometer YES UVB-1 at OAPA. White line and grey area represent the climatological UVI at noon in Río Gallegos.**

| 675K | 950K |
| :---: | :---: |

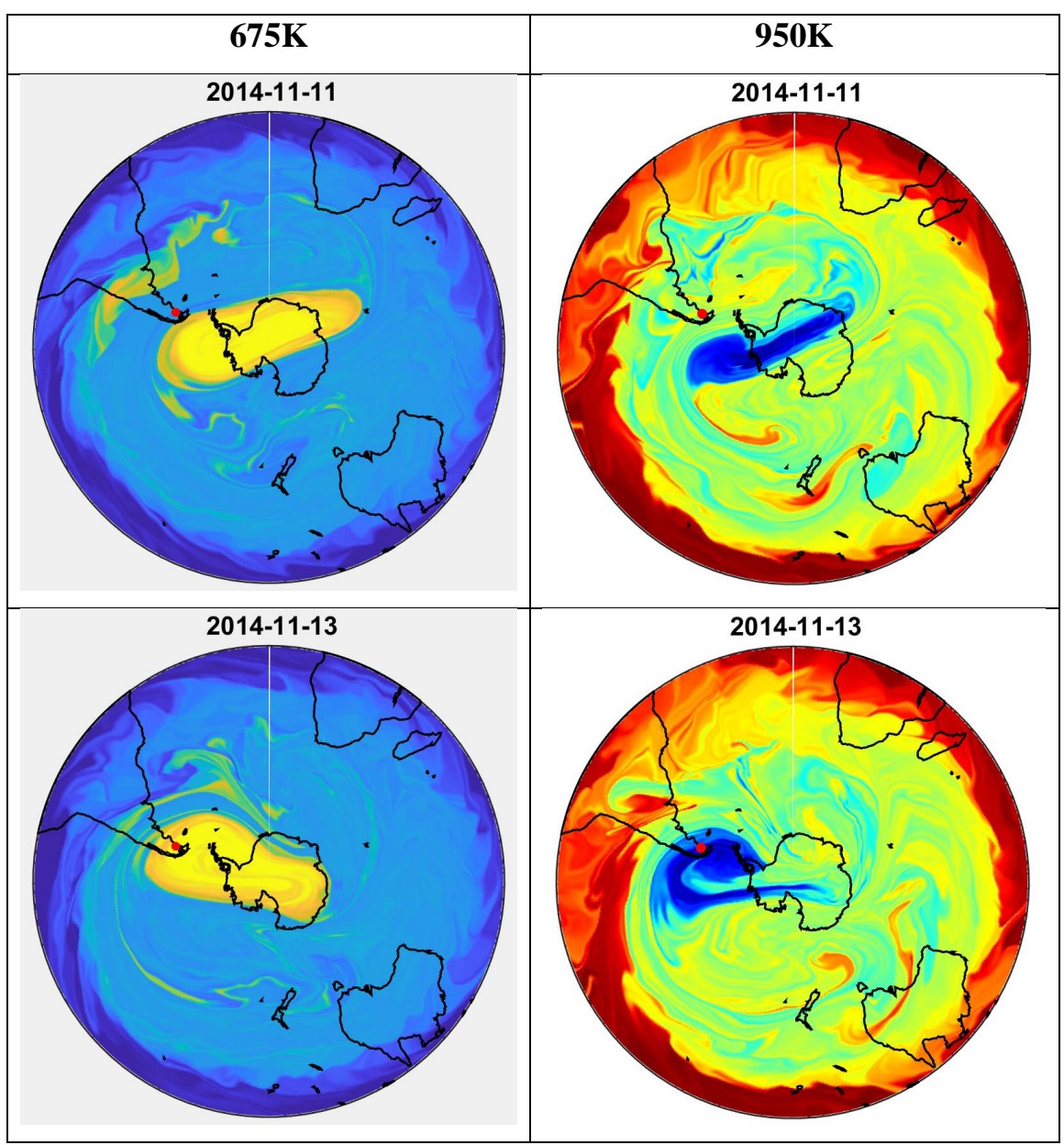

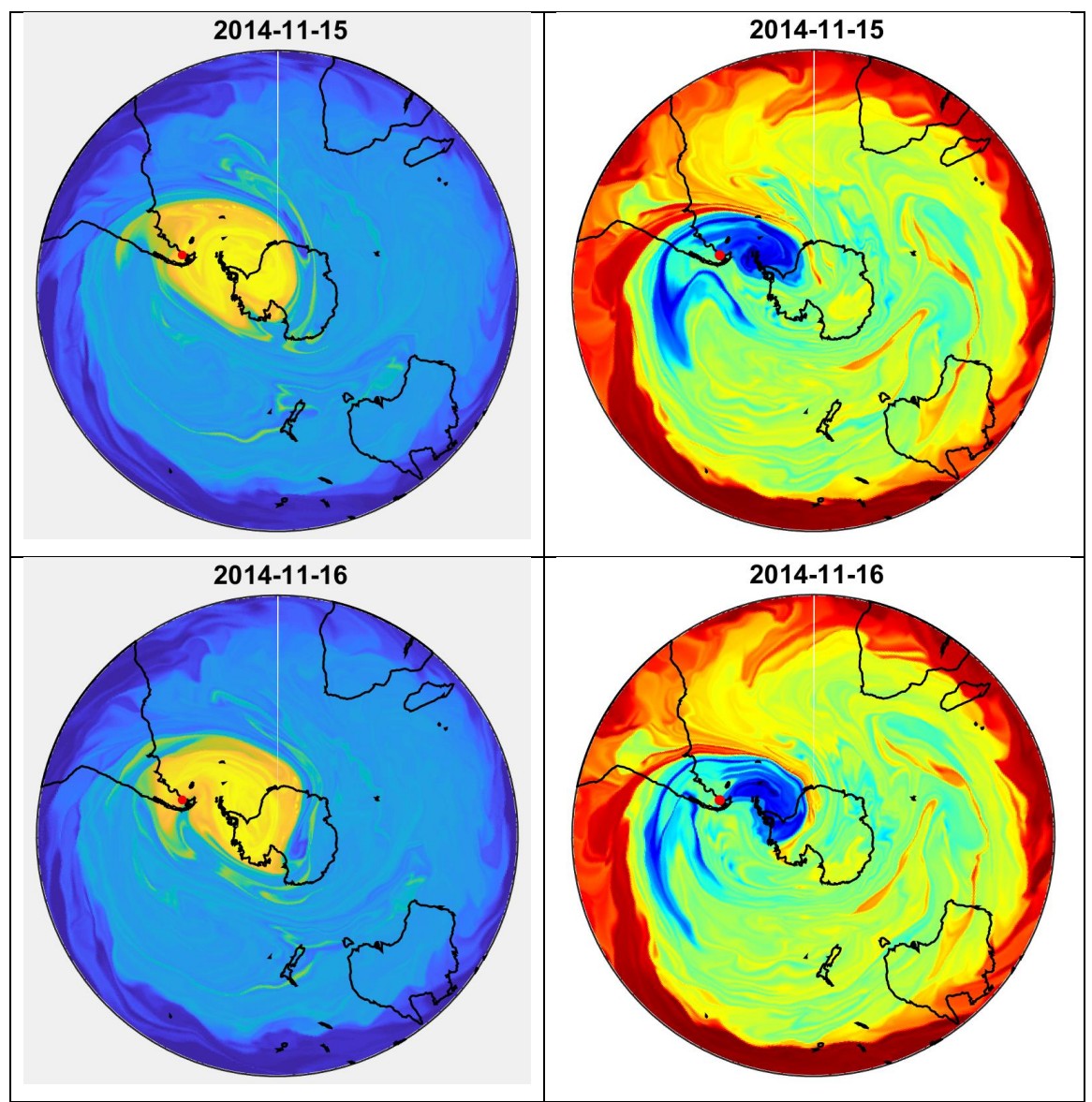

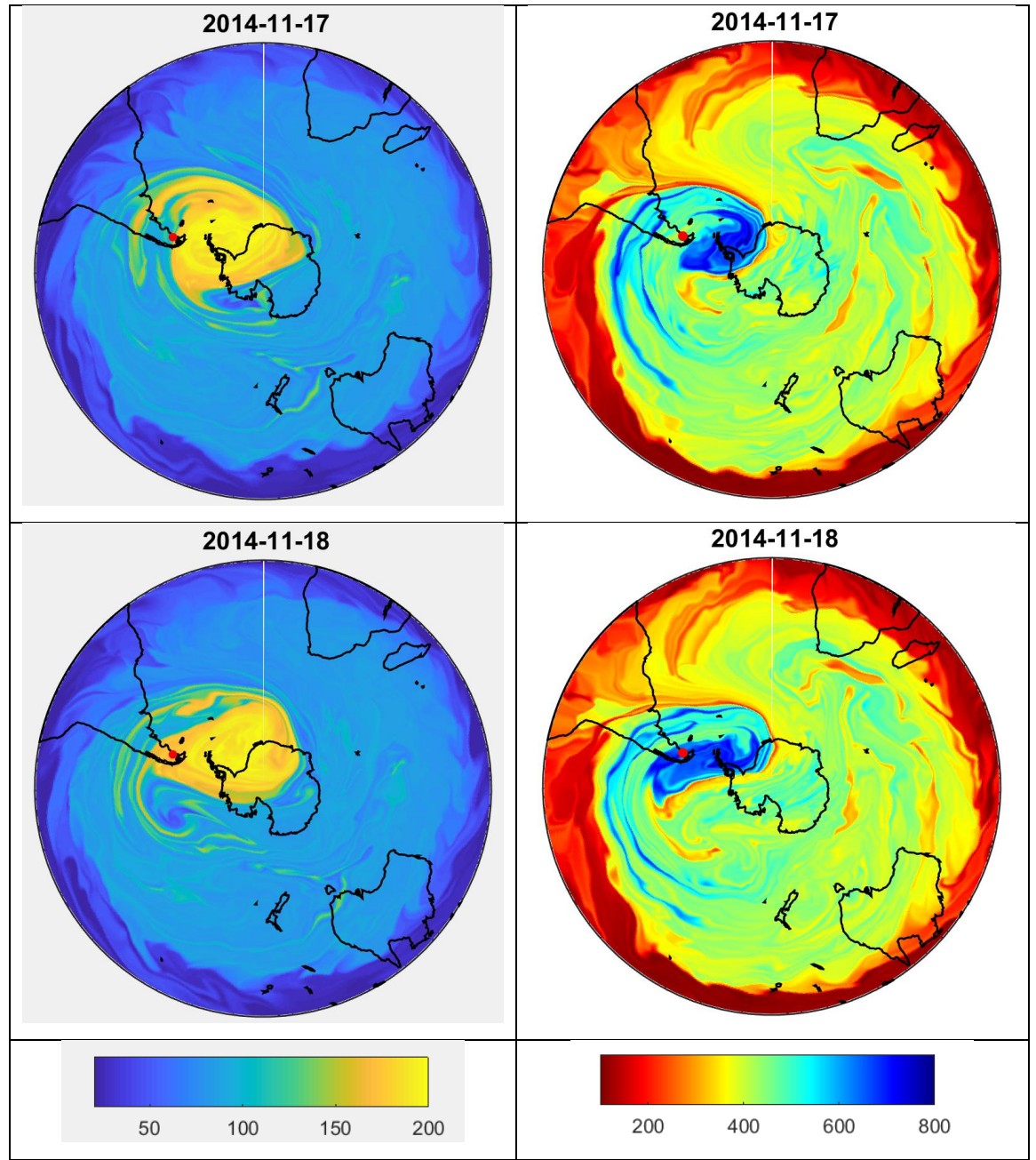

**Figure 8 Advected Potential Vorticity maps assimilated with the MIMOSA model. Maps show the evolution of the polar vortex for two isentropic levels with potential temperatures of 675K (left) and 950K (right).**

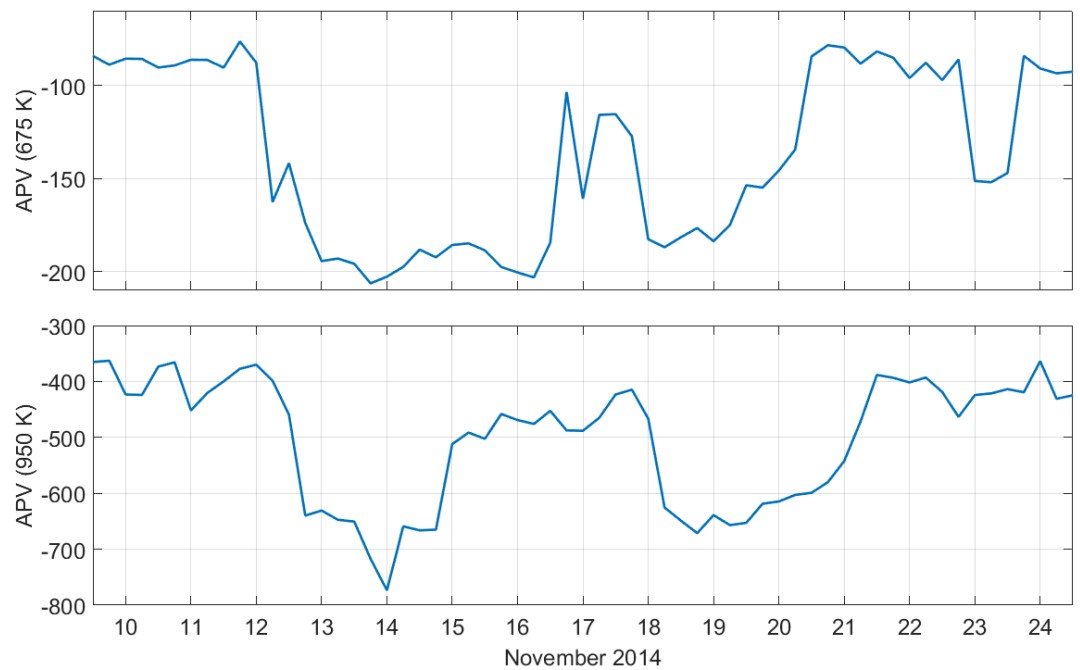

**Figure 9 Time evolution of Advected Potential Vorticity assimilated with the MIMOSA model over Río Gallegos (-51.6; -69.3) at 675 K (top) and 950 K (bottom)**