# Peer review of "Analysis of a southern sub-polar short-term ozone variation event using a Millimeter-Wave Radiometer"

_Annales Geophysicae, 2019_

## Referee Comment (RC1) · Lucas Vaz Peres (Referee) · 23 Mar 2019

Orte et al., 2019:

"Analysis of an event of short term ozone variation using a Millimiter - Wave Radiometer installed in subpolar region".

General comments:

The authors present a study about an atypical event of polar vortex and ozone hole influence over Río Gallegos during November of 2014. This event was detected from the Millimeter Wave Radiometer (MWR) measurements at 27 and 37 km and the advected

potential vorticity (APV) was calculated from the high-resolution advection model MI-MOSA (Modélisation Isentrope du transport Mésoéchelle de l'Ozone Stratosphérique par Advection) at 675 and 950 K to understand and explain the atmospheric dynamic related to ozone rapid variation during the passage of the polar vortex. In addition, the MWR dataset were compared for first time with Microwave Limb Sounder (MLS) to 27 km, 37 km and 65 km and with the Differential Absorption Lidar (DIAL) installed in Observatorio Atmosférico de la Patagonia Austral (OAPA) between October 2014 and 2015. This work is a useful representation of the important contribution made by the Millimeter Wave Radiometer (MWR) at Río Gallegos and certainly, understand the ozone hole influence over Río Gallegos is of fundamental importance in many environmental processes which can lead to increases in the UV radiation on the surface. This increase in the UV radiation related to ozone reductions can be dangerous to life on earth and it represent a significant scientific advance. It should be published after some modification as present clearly objectives, if it's the comparison multi-instrument or the ozone reduction study case and precisely discuss the results with the literature (This is the worst article failure). I believe there was a mistake in section 5 Discussion. Because of these I would recommend to accept with Major Revision this manuscript.

Specific comments:

ïČŸ In the abstract: The abstract must clearly highlight the most significant scientific result, besides first present the main results of comparison between the data sets and after explain the occurrence of the event.

ïČŸ The 1. Introduction have a good structure but needs to improve the "historical" contextualization of the scientific problem "ozone transport", documenting it better in the literature.

WAUGH 1993; can be of great value to help in the contextualization of the subject, since that, indirectly, the Ozone Hole can influence the ozone content of medium- and low-latitude regions through the release of polar filaments, which carry air masses of

ozone-depleted from the Antarctic polar vortex, causing a temporary decrease in the total ozone column over these regions.

WAUGH, D. W. Subtropical stratospheric mixing linked to disturbances in the polar vortices. Nature, v. 365, p. 535–537, 1993.

Moreover, KOCH et al., 2002 explain that the extreme anomalies in the total ozone content in mid-latitudes of the stratosphere are associated with the southern transport of regions where the climatological concentrations are lower or higher.

KOCH, G.; WERNLI, H.; STAEHELIN, J.; PETER, T. A Lagrangian analysis of stratospheric ozone variability and long-term trends above Payerne (Switzerland) during 1970–2001. J. Geophys. Res., v. 107, n. D19, p. ACL 2-1–ACL 2-14, 2002.

Objectives should highlight the scientific advance that the article want produces

- Pg 2, line 6. Missing reference in this sentence. - Pg 2, line 8. Short paragraph, may be part of the previous paragraph. - Pg 2, line 12. Missing reference in this sentence. - Pg 2, line 17. In the sentence "Its will remain for decades in the atmosphere, destroying ozone on the Antarctic pole" the Artic pole can be inserted. - Pg 3, lines 8 - 10. "The transport of polar air masses may take the form of "filaments" and "tongue", which induce anomalies on the ozone and UV observations over mid-latitudes": Define filament and language in literature. Referring the paragraph in the literature. - Pg 3, line 12. Short paragraph, may be part of the previous paragraph. - Pg 3, lines 21 - 28. This paragraph seems to me to be better positioned in the methodology.

ïČŸ In the 2. Materials and methods:

- 2.1.1 Pg 5, line 3. Define "Glass Dewar".

Pg 5, lines 5 – 6. Short paragraph, may be part of the previous paragraph.

- 2.2 It is necessary to show the potential vorticity equation and their terms.

Define filaments and tongues observed in the MIMOSA PV fields.

- 2.3 What is the criterion used to identify the occurrence of the polar vortex and ozone hole influence over Río Gallegos? Reduction in ozone and PV values? which? About what?

Pg 9, line 2: How was the opacity calculated?

Explain better why the heights of 27, 37 and 65 km were chosen to make the comparison. ïČŸ In the 3. Inter-comparison of MWR with DIAL system and MLS observations

- This section should be within the results

- Pg 9, line 18. Check the figure number. I think this is 3.1.

- "This is because the DIAL measurement campaign becomes more intense in those months when the ozone hole approaches southern Argentina." should be replaced by: "This is because the DIAL measurement campaign becomes more intense in those months when the ozone hole is active and approaches over the southern Argentina". Referring the sentence in the literature.

- 3.1 Figure 3.1 should be 3.2.

Values of tables 3.1 and 3.2 may be in said figures in order to optimize space.

What criteria are used to call the correlations of considerable (pg 10, line 6), acceptable (pg 10, line 9 and pg 10, line 23), moderate (pg 10, line 10), and very good (pg 10, line 13)?

"The MBE was calculated to analyse the bias between satellite and ground-based data. We obtained a value of +5% indicating an MWR overestimation with respect to the MLS". Validation is usually done from satellite equipment in relation to ground-based equipment, not the reverse as was done here.

Pg 10, line 9. 11% difference is reliable in the literature.

Pg 10, line 12. Which represent the slope and intercept values?

The results of this section should be discussed in the literature. This is a major flaw of this article.

- 3.2 Figure 3.5 should be 3.3. The results of this section should be discussed in the literature. This is a major flaw of this article.

ïČŸ In the 4. Results

- 4.1, 4.2 and 4.3. Results are well described but need to be discussed in the literature.

- 4.2. Remove "trend" in pg. 11, line 12 and 21. If you use this term you need to explain how the trend was calculated

ïČŸ In the 5. Discussion

- This section, in my opinion, should not exist. The results should be discussed as they are described. - I as a reader was anxious for discussion in literature, but I had an unpleasant surprise at seeing only one reference. The way it is, it's not a discussion.

- Much of what is written in this section can enrich the conclusions.

ïČŸ Conclusions - What scientific progress was made in the study? - As tip I suggest to merge what is written in the "Discussion".

ïČŸ References - Put in alphabetical order.

ïČŸ Figures:

- Figure 2.2. Explain in the text why MWR fall data between March and April and July and August. - Figure 5.1 should be in the methodology

Please also note the supplement to this comment:
https://www.ann-geophys-discuss.net/angeo-2019-17/angeo-2019-17-RC1-supplement.pdf

─────────────────────

2019.

---

## Referee Comment (RC2) · Anonymous Referee #2 · 1 Apr 2019

Journal: Annales Geophysicae

Revision of MS No.: angeo-2019-17

Title: Analysis of an event of short term ozone variation using a Millimiter-Wave Radiometer installed in subpolar region.

Authors: Pablo Facundo Orte, Elian Wolfram, Jacobo Salvador, Akira Mizuno, Nelson Bègue, Hassan Bencherif, Juan Lucas Bali, Raúl D'Elia, Andrea Pazmiño, Sophie Godin-Beekmann, Hirofumi Ohyama, Jonathan Quiroga.

Overall evaluation

[Figure]

This manuscript analyses an event of polar-vortex-related unusual ozone decreasing at height levels of 27 and 37 km on the stratosphere over Río Gallegos, Argentina, during November of 2014, through a set of remote ground and satellite measurements and dynamical modelling. The subject is appropriate for the scope of Annales Geophysicae. The multiple tools used to analyse the event, and their intercomparison, gives robustness to the work. Results and conclusions imply in general a relevant contribution to the field, given that this type of localized sub-polar ozone reductions, and eventual "mini ozone holes" at lower latitudes, is an atmospheric subject by itself. There are, however, several aspects to revise in order to put the manuscript in conditions to be accepted for publication.

Specific comments:

- The manuscript's title must be as concise and direct as possible, emphasizing the object of study instead one of the used tools. I suggest some like: "Analysis of a November 2014 southern sub-polar short-term ozone variation event". Eventually, if the MWR instrument is cited, please change "Millimiter" by "Millimeter".

- A conceptual aspect to revise throughout the manuscript is the coherence and rigor in the use of terms "polar vortex" and "ozone hole". The "Antarctic polar vortex" is a dynamical phenomenon which has been present probably for millions of years, and their mention is essential when the dynamics is analyzed particularly as a function of the altitude. While, the "Antarctic ozone hole" is the extreme manifestation of the stratospheric ozone layer depletion in the interior of the "Antarctic polar vortex", which has made evident since late 1970s, and is mainly referred to either when their vertical ozone structure is afforded or their consequences on surface are analysed. To speak of "ozone hole", for definition the vertical total ozone column values must fall below 220 DU; authors must revise their use when appropriate. In turn, terms as "ozone hole influence" are appropriate for sub-polar regions but in this case explicit mention to the "Antarctic ozone hole" must be made, eventually an abbreviation AOH may be useful. Similarly, phrases as (page 12, lines 19-20) "the southern part of South America has

been affected by the systematic and abrupt intrusion of the polar vortex during the spring since the 1980's" are inappropriate: as said, the Antarctic polar vortex occurs probably since millions years ago, the difference is that before the 1980s their interior produced no "ozone hole", i.e. ozone values below 220 DU as it is defined, and without the presence of the ozone hole probably the polar vortex intrusions would have no major transcendence for the surface. Authors must take particular care about the use of these key expressions. In this phrase, also the word "systematic" is inappropriate. It could be changed by some like: "the southern part of South America has been affected by the frequent abrupt intrusions of the AOH during the spring since the 1980's". Similarly, the phrase (page 11, line 14) "This decrease is related to the passage of the ozone hole over Rio Gallegos" is wrong, as TOC never falls below 220 DU. Several other paragraphs along the manuscript must be revised accordingly.

- Given that the vertical total ozone column (TOC) values are a necessary reference when ozone anomalies are reported, I suggest a detailed mention to the TOC not only when the present case is analysed but also when mention to other cited cases to help distinguish Antarctic ozone hole "influences" from Antarctic ozone hole "overpass", and ozone hole "reductions" from eventual "mini ozone-holes" or real ozone hole "overpass".

- In the Introduction: as a benchmark for the specific analysis of this work, it would have been desirable a characterization, based on references, of the known springtime typical vertical structure of the atmosphere over southern South America on both "sides" (inner/outer) of the Antarctic ozone hole.

- In the same sense, specific parts of these references could be useful to compare and put in major context the results from this work.

- Given that one of the concerns with ozone negative anomalies is the potential increase in harmful UVB solar irradiance at ground, please could you add, e.g. in Figure 6, other plot of locally-measured clear-sky UV Index (at noon, or at a given fixed solar

zenith angle) allowing quantify the simultaneous UVB increase for these days?.

Minor comments:

Text

- Please define the abbreviations the first time the parameters are mentioned, and then use just the abbreviation. E.g. page 11, line 14: standard deviation is mentioned before, abbreviation SD should be presented the first time it is mentioned and then only SD used. The same for TOC in line 18.

- Page 3, lines 1-2: please change by "due mainly to tropospheric-stratospheric dynamical processes".

- Page 3, lines 6-10: I think a change in the order of paragraphs would make more coherent this sentence. I suggest: "The transport of polar air masses may take the form of "filaments" and "tongue", which induce anomalies on the ozone and UV observations over mid-latitudes. Recently, based on satellite and ground-based observations in Uruguay and Southern Brazil, Bresciani et al. (2018) showed a decrease of ozone over these sites during October 2016 in link to this phenomenon".

- Page 3, line 9: "which induce anomalies on the ozone and UV observations". Anomalies are on the ozone and UV behavior, not on the observations. Please correct.

- Page 3, lines 22-23: phrase "The OAPA is located in sub-polar latitudes, which makes it a suitable site to study stratospheric ozone due to its closeness to the Antarctic ozone hole" is wrong. It could be: "The geographical location of OAPA makes it a suitable site to study the sub-polar stratospheric ozone due to its closeness to Antarctica".

- Page 3, line 31: "decreasing the ozone amount" instead "increasing the ozone amount"?.

- Page 4, some paragraphs of lines 1 up to 8 seem more appropriate for section 2. Materials and Methodology, other for the conclusions and future possibilities. Please

redistribute them.

- Page 7, line 11: define AMF.

- Page 8, line 3: "into the daily cycle": did you mean "within the diurnal cycle"?. In line 4: please rewrite "that this gas suffer in this layer" in other form.

- Page 9, line 22: replace "Argentina" by "South America".

Figures

- Text of Page 11, line 10: . . . "light red". . . but in the caption of Figure 6 it is referred to as "pink".

- Figure 9 and several paragraphs from the Introduction treating on the characteristics of the measurement site (e.g. page 3, lines 21 on) should be at the start of section 2. Materials and Methodology.

- The captions of the figures must contain all the information needed to interpret them. Please revise the captions of all figures. In Figure 2 please correct . . . ratio for three altitudes: 27, 37 and 65 km.

- The abscissas and ordinates legends and labels must explicit clearly the parameters in each axis. E.g. in Figures 2, 4 and 5, the y-legends must include "ozone mixing ratio". Dates in Figure 3 are better presented in Figure 2. In Figures 4, 5 the altitude may be in form of title for each plot. In Figure 6 the year is not specified, don't use the abbreviation TOC.

These comments may be considered as relatively "minor changes". However, I suggest they should be taken as mandatory for a posterior re-evaluation of the manuscript.

---

## Author Comment (AC1) · 25 Apr 2019

The authors acknowledge to Prof. Lucas Vaz Peres and the editor for the time spent to review this manuscript and also for their constructive comments. The manuscript was revised and improving according the referee comments and suggestions.

(A clearer .pdf file is upload as supplement file)

Answers to the referees comment's:

Specificy comments:

In the abstract:

[Figure]

RC: The abstract must clearly highlight the most significant scientific result, be­sides first present the main results of comparison between the data sets and after explain the occurrence of the event.

AC: To highlight the results of the study case in the abstract it was improved and the following sentence was inserted in the abstract (marked with bold) (Pg. 1, lines 25-26 and 29-30):

"The measurement shows a very short term recovery in the middle of ozone mixing ra­tio decrease that could be detected by the MWR. The advected potential vorticity (APV) calculated from the high-resolution advection model MIMOSA (Modélisation Isentrope du transport Méso-échelle de l'Ozone Stratosphérique par Advection) was also anal­ysed at 675 and 950 K to understand and explain the dynamic at both altitudes and correlate the ozone rapid recovery measured with the passage of a filament with higher AVP values over Río Gallegos."

The 1. Introduction:

RC: Have a good structure but needs to improve the "historical" contextualization of the scientific problem "ozone transport", documenting it better in the literature. WAUGH 1993; can be of great value to help in the contextualization of the subject, since that, indirectly, the Ozone Hole can influence the ozone content of medium- and low­latitude regions through the release of polar filaments, which carry air masses of ozone-depleted from the Antarctic polar vortex, causing a temporary decrease in the total ozone column over these regions. WAUGH, D. W. Subtropical stratospheric mix­ing linked to disturbances in the polar vortices. Nature, v. 365, p. 535–537, 1993. Moreover, KOCH et al., 2002 explain that the extreme anomalies in the total ozone content in mid-latitudes of the stratosphere are associated with the southern transport of regions where the climatological concentrations are lower or higher. KOCH, G.; WERNLI, H.; STAEHELIN, J.; PETER, T. A Lagrangian analysis of stratospheric ozone variability and long-term trends above Payerne (Switzerland) during 1970–2001. J.

Geophys. Res., v. 107, n. D19, p. ACL 2-1–ACL 2-14, 2002.

AC: The introduction is modified (pg. 3, line 19-28) and we add the proposed reference (WAUGH 1993), as follow:

"The air-mass transport in the stratosphere has been extensively analysed using the advected potential vorticity (APV) which is considered a suitable dynamical tracer in the stratosphere. The transport of polar air masses may take the form of "filaments" or "tongue". These terms had been used to explain the transport of air from the edge of the polar vortex into middle latitudes by Waugh (1993) analysing potential vorticity maps, and previously, to explain the intrusion of tropical air into mid-latitudes by Randal et al. (1993). When the intrusion of air from the polar vortex reaches mid-latitudes and produce ozone decreases, it induces anomalies on the surface UV radiation. Bittencourt et al. (2018) also linked the occurrence of this event over South America to later changes in the tropospheric and stratospheric dynamic behaviour. Thus, this parameter can be used to study the dynamics of the Antarctic polar vortex and as a tracer of poor-ozone air masses that are released from the ozone hole (Bittencourt et al., 2018; Kirchhoff et al., 1996, Pinheiro et al., 2011; Wolfram et al., 2012; Hauchecorne et al., 2002; Marchand et al., 2005; Bencherif et al., 2007)."

RC: Objectives should highlight the scientiïfic advance that the article want produces

AC: Paragraph from pg. 3 line 29 to pg. 4 line 5 was improved aiming in highlight the main objectives of this work: Describe the study case, present the capability of the MWR regarding with the high temporal resolution that allows to study short-term ozone variation and present for the first time an inter-comparison with independent ground-based and satellite instruments. These objectives are then reinforced in conclusion section. The paragraph was modified as following: "In this paper we analyse an unusual event of rapid decrease and recovery of volume mixing ratio over Río Gallegos, Argentina, during November 2014 due to the release of a tongue of a poor-ozone air mass. This analysis was achieved by means of ground and space-based instruments,

focusing on the MWR ozone measurements. The high temporal resolution (one hour) of the MWR observations are analysed at different altitudes (27 and 37 km) with the aim to determine the short-term variability of ozone mixing ratio and the moment when the polar vortex and its edge (as tongue or filamentary structure) with poor-ozone air masses pass over Río Gallegos and leave it at those altitudes, resulting in a local peak of ozone mixing ratio for a very short period of time on November 2014. TOC measurements are also analysed by the ground-based instrument SAOZ installed in OAPA and by the satellite Ozone Monitoring Instrument (OMI). Finally, the APV field from the MIMOSA model was used to analyse the air-mass transport during the event. In addition, the MWR ozone mixing ratio retrieved in Río Gallegos is compared for the first time with ground-based measurements from the ozone DIAL/NDACC instrument and satellite measurements from the MLS on board the AURA/NASA."

RC: Pg 2, line 6. Missing reference in this sentence.

AC: Reference was added in the revised manuscript (Pg. 2, line 6): "Without atmospheric ozone, life would not be possible as we know it today. Although most production takes place in the equatorial region due to the higher level of solar radiation, the maximum ozone concentration is observed over the polar region (Salby, 1996)."

Reference: Salby, M. L.: Fund, Atmos. Phys. International geophysics series, Academic Press, Vol. 61, 1996.

RC: Pg 2, line 8. Short paragraph, may be part of the previous paragraph.

AC: Short paragraph was added to the previous paragraph.

RC: Pg 2, line 12. Missing reference in this sentence.

AC: Reference was added (Pg. 2, line 11).

"This ozone destruction is the consequence of human emission of components containing chlorine and bromine into the atmosphere, called Ozone Depleting Substances (ODS) (WMO, 2011)."

Reference: World Meteorological Organization (WMO): ScientiïñĄc Assessment of Ozone Depletion: 2010, Global Ozone Research and Monitoring Project-Report No. 52, 516 pp., Geneva, Switzerland, 2011.

RC: Pg 2, line 17. In the sentence "Its will remain for decades in the atmosphere, destroying ozone on the Antarctic pole" the Artic pole can be inserted.

AC: We agree with this comment, but we want to emphasize on the destruction of the Antarctic ozone which is the phenomenon that is involved in the case study proposed in the manuscript and it is the most important in terms of ozone destruction amount. We decided to include the word "mainly" into the sentence with the intention to reflect that the Antarctic pole is not the only place where ozone destruction may take place, but it is the strongest destruction (pg. 2, line 16-18).

"However, the lifetime of these compounds in the atmosphere is very long (e.g, 100 years for some of them) (M. Rigby et al., 2013, 2014; WMO, 2014) and it will remain for decades in the atmosphere, destroying ozone mainly over the Antarctic polar region."

RC: Pg 3, lines 8 10. "The transport of polar air masses may take the form of "ïñĄlaments" and "tongue", which induce anomalies on the ozone and UV observations over mid-latitudes": DeïñĄne ïñĄlament and language in literature. Referring the paragraph in the literature.

AC: This is modified as described above (pg. 3, line 19).

RC: Pg 3, line 12. Short paragraph, may be part of the previous paragraph.

AC: Short paragraph was added to the previous paragraph, as suggested.

RC: Pg 3, lines 21 - 28. This paragraph seems to me to be better positioned in the methodology.

AC: The paragraph was improved and moved to section 2 (Materials and Methodology).

In the 2. Materials and methods:

RC: 2.1.1 Pg 5, line 3. Define "Glass Dewar".

AC: The definition of "Glass Dewar" was added (pg.5, line 17).

"The hot blackbody load is achieved using a radio absorber at room temperature ($\sim$300 K), while the cold load is achieved by soaking a similar absorber in Liquid nitrogen (77 K) contained in a glass Dewar (vacuum bottle made of glass that is used especially for storing liquefied gases)."

RC: Pg 5, lines 5 – 6. Short paragraph, may be part of the previous paragraph.

AC: Short paragraph was added to the previous paragraph.

RC: 2.2 It is necessary to show the potential vorticity equation and their terms. Define filaments and tongues observed in the MIMOSA PV fields.

AC: Instead of including the APV equation, reference containing full description of the MIMOSA PV calculation was included (pg 8, line 5).

Reference: Heese, B., S. Godin, and A. Hauchecorne, Forecast and simulation of stratospheric ozone filaments: A validation of a high-resolution potential vorticity advection model by airborne ozone lidar measurements in winter 1998/1999, J. Geophys. Res., 106 (D17), 20011-20024, 2001.

RC: 2.3 What is the criterion used to identify the occurrence of the polar vortex and ozone hole influence over Río Gallegos? Reduction in ozone and PV values? which? About what?

AC: "Ozone hole influence" is used when the ozone hole is not over Río Gallegos, but there are ozone amount reduction as consequence of the formation of the ozone hole over the Antarctic. The ozone hole is defined by reduction of total ozone column below to 220DU. The identification of the polar vortex (or edge, or filamentary structure or tongue) is obtained by analyzing the APV. To clarify this point, we modified a paragraph in section 1 (Introduction) (pg. 3, lines 9-13) and the text was reviewed. Specifically

talking about the case study of short-term ozone variation, firstly it is determined the case of rapid variation (decrease or increase) in the ozone mixing ratio by mean of the MWR measurements. Then, analyzing the APV, it is confirmed that the air mases with poor ozone masses are coming from the edge of the polar vortex.

RC: Pg 9, line 2: How was the opacity calculated?

AC: The Opacity Observations is obtained as following:

The radiative transfer equation for microwave remote sensing considering a non scattering and an isothermal medium, can be written as follow (Janssen, 1993):

$T\_b = T \int \_@(\tau\_a) ▒e^(-\tau) d\tau = T(1 - e^(-\tau\_a))$ where T_b is the brightness temperature, T is the temperature of the source, and $\tau$_a is the total optical thickness in the optical path of radiation propagation. In the problems of remote sensing of the atmosphere, T_b is generally obtained through the measurement and it is desired to infer some component or atmospheric property such as the distribution of ozone for our case, water vapor or temperature.

The observations of the middle-atmosphere with remote sensing techniques from the ground, suffer the extinction of the atmospheric layers that are below, mainly for the troposphere. In the range of micrometer waves, scattering can be neglected. Absorption is produced primarily by water vapor and to a lesser extent, oxygen and other gases. These gases are concentrated in the first kilometers of the atmosphere. Also they emit radiation in the frequency range of measurement, known as continuous emission (if no discrete absorption pick is near to the frequency analyzed). If we turn away in frequency from the characteristic ozone emission line in the measured radiation spectrum, only we have the contribution of the continuous emission from the troposphere and the absorption can be described by the Beer-Lambert law. Thus, assuming an isotherm troposphere, we can adapt the previous equation to describe the signal from the lower atmosphere for a given angle of observation as:

[Figure]

T_(b_low )=T_trop (1-e^(- $\tau$_z/cos($\theta$_low ) ) )+T_sys

Where T_(b_low ) is the brightness temperature observed by the MWR at $\theta$_low, T_trop is the average temperature of the troposphere, $\tau$_z is the zenith opacity, $\theta$_low the zenith angle of observation and T_sys is the term that describes the instrumental noise. On the other hand, the signal from the hot black body at room temperature will be:

T_hot=T_hot'+ T_sys

Where T_hot' is the signal from the hot source in brightness temperature units, without the contribution of system noise. Differentiating these two signals, assuming T_trop=T_hot' and applying natural logarithm on both sides, we have:

ln⁡(T_hot-T_low )=ln⁡ãĂŰãĂŰ(TãĂŮ_hot')-$\tau$_z/cos($\theta$_low ) ãĂŮ

This equation describes a linear relation between the secant of the zenith angle and ln⁡ ln⁡(T_hot-T_low ) with slope $\tau$_z and intercept ãĂŰln⁡ãĂŰ(TãĂŮãĂŮ_hot'), which is considered equal to ln⁡ãĂŰãĂŰ(TãĂŮ_trop)ãĂŮ. Therefore, plotting these observations measure at different directions (Figure below), on the axis x -1/cos⁡($\theta$) and y axis as ln⁡(T_hot-T_low), $\tau$_z and T_trop can be obtained through a linear fit (red line). In this example, an opacity of 0.283 and T_trop=407.483 is obtained. This method is known as "tipping-curve".

Figure. Example of the retrieval of opacity from MWR observations. —-o—- In the manuscript, we add a sentence mentioning that the opacity is retrieved from the MWR and the reference that describe the procedure to obtain the opacity. The following sentence is added in the manuscript (Pg. 9, line 19): "The opacity is retrieved by the MWR during the measurement cycle (Orte, 2017)." The full description of the procedure to obtain the opacity can be found in the reference Orte, 2017, pg.62. It can be found at the following link: http://ria.utn.edu.ar/handle/123456789/20, 2017. Reference: Orte, P. F.: Procesamiento de señales de un radiómetro de ondas milimétricas para obtener

perfiles de ozono y estudios de la radiación solar UV en superficie, PhD Thesis, UTN-FRBA,http://ria.utn.edu.ar/handle/123456789/20, 2017

RC: Explain better why the heights of 27, 37 and 65 km were chosen to make the comparison.

AC: It was added in subsection 2.3 (Methodological considerations), pg. 8 line 14-20.

In the 3. Inter-comparison of MWR with DIAL system and MLS observations

RC: This section should be within the results

AC: The section "Inter-comparison of MWR with DIAL system and MLS observations" is now adapted and moved to the Result section (please see revised manuscript).

RC: Pg 9, line 18. Check the figure number. I think this is 3.1.

AC: In Pg 9, line 18 it is mentioned the Figure 3, which is consistent with the text. As there is not figure 3.1, the agreement of all figure numbers were checked to corroborate the consistence in the text.

RC: "This is because the DIAL measurement campaign becomes more intense in those months when the ozone hole approaches southern Argentina." should be replaced by: "This is because the DIAL measurement campaign becomes more intense in those months when the ozone hole is active and approaches over the southern Argentina". Referring the sentence in the literature.

AC: It is replaced by (pg. 10, line 9):

"This is because the DIAL measurement campaign becomes more intense in those months when the ozone hole approaches and overpasses the southern Argentina (Wolfram et al., 2012)."

Reference: Wolfram, E. A., Salvador, J., Orte, F., D'Elia, R., Godin-Beekmann, S., Kuttippurath, J., Pazmiño, A., Goutail, F., Casiccia, C., Zamorano, F., Paes Leme, N.,

and Quel, E. J.: The unusual persistence of an ozone hole over a southern mid-latitude station during the Antarctic spring 2009: a multi-instrument study, Ann. Geophys., 30, 1435-1449, https://doi.org/10.5194/angeo-30-1435-2012, 2012.

RC: 3.1 Figure 3.1 should be 3.2.

AC: As there is no figure 3.1 and 3.2, the agreements of all figure numbers were checked to corroborate the consistence in the text.

RC: Values of tables 3.1 and 3.2 may be in said figures in order to optimize space.

AC: The values of both tables were merged in one table as following (pg. 27). The text was adapted to the new table (please, see load figure 1):

Alt. N Slope Intercept [vmr(ppm)] R MBE MWR-MLS 27 km 84 1.01 0.24 0.65 +5% 37 km 84 0.96 -0.43 0.63 -11% 65 km 84 0.95 0.02 0.88 -7% MWR-DIAL 27 km 30 0.93 0.36 0.73 -1%

RC: What criteria are used to call the correlations of considerable (pg 10, line 6), acceptable (pg 10, line 9 and pg 10, line 23), moderate (pg 10, line 10), and very good (pg 10, line 13)?

AC: The criterion used for these words was made taking into consideration the closeness of the correlation coefficient to one. A perfect positive linear correlation is when this value equals one. We decided to remove the words "acceptable" and "moderate" with the aim to reduce the subjectivity. The paragraph mentioned is modified as follow:

"Unlike the average ozone mixing ratio at 27 km, the MBE at 37 km reflected an underestimation of ozone mixing ratio of -11% compared with MLS. Fiorucci et al. (2013) also presented differences ranging between -8% and -18 % in the 17–50 km vertical range, reaching ∼-18% at 37 km. The regression analysis presents a slope of 0.96 and an intercept of 0.44. Similarly, the correlation coefficient at this altitude was calculated (R = 0.63) to evaluate the correlation between MWR and MLS at this altitude." RC: "The MBE was calculated to analyse the bias between satellite and ground-based data. We

obtained a value of +5% indicating an MWR overestimation with respect to the MLS". Validation is usually done from satellite equipment in relation to ground-based equipment, not the reverse as was done here.

AC: The justification of calculate the MBE in this way is that we realize comparisons between measurements with the aim to determine the bias of the MWR respect the MLS and DIAL with the intention that the positive sign reflect an overestimation of the instrument under analysis (MWR) respect others independent instruments (DIAL and MLS), while a negative sign reflecting underestimation. Similar comparisons between these types of instruments where the MWR is analysed in relation of satellite instruments can be found, for example, in Ohyama et al. 2016 or Schneither et al. 2003, among others.

RC: Pg 10, line 9. 11% difference is reliable in the literature.

AC: In literature can be found bigger differences. For example, Fioruchi et al. (2013) reported a difference around 18% at 37 km (from Figure 3), with differences ranging between 8 % to 18 % in the 17–50 km vertical range. Discussion in the literature was added in Discussion section.

RC: Pg 10, line 12. Which represent the slope and intercept values?

AC: In this case, the linear regression is used to evaluate the comparison. The slope represents how much increase (or decrease) the MWR measurement when the "control" (MLS) measurement increases (or decreases), plus or minus the uncertainty values. As we have the same desired quantity measured by both instruments intercompared, the optimal slope will be one. It would indicate that changes in the reliable measurements from the "control" instrument have the same change as the measurements retrieved from the instrument under analysis, plus a random error. The word "control" is used here to refer the validated instrument (MLS) as a reference. The same can be said in regard to the intercept estimation. An "optimal" value for the intercept would be 0, indicating no bias from the MWR instrument compared to the reference

one (MLS).

RC: The results of this section should be discussed in the literature. This is a major flaw of this article. - 3.2 Figure 3.5 should be 3.3. The results of this section should be discussed in the literature. This is a major flaw of this article.

AC: There were added the discussion in the literature to evaluate our results in term of the consistence with other results. In addition, the Discussion section (section 4) was improved as we detail below.

In the 4. Results

RC: 4.1, 4.2 and 4.3. Results are well described but need to be discussed in the literature.

AC: The Discussion section (section 4) was improved in this way. Please, see this section in the revised manuscript.

RC: 4.2. Remove "trend" in pg. 11, line 12 and 21. If you use this term you need to explain how the trend was calculated.

AC: The word "trend" was removed. The phrases were replaced as follow:

Pg.11. line 28: The phrase "We observe a rapid ozone decrease trend at both altitudes from November 11 at 19:30 local time (LT) to November 15" was replaced by "We observe a rapid ozone decrease at both altitudes from November 11 at 19:30 local time (LT) to November 15"

Pg. 12, line 6: The phrase "The general trend of both measurements follows the behaviour of the MWR at 27 and 37 km and it shows the influence of the ozone hole..." was replaced by "The general behaviour of both measurements follows the behaviour of the MWR at 27 and 37 km and it shows the influence ..."

In the 5. Discussion

RC: This section, in my opinion, should not exist. The results should be discussed as they are described. - I as a reader was anxious for discussion in literature, but I had an unpleasant surprise at seeing only one reference. The way it is, it's not a discussion. - Much of what is written in this section can enrich the conclusions.

AC: Attending to this important comment, the discussion section was improved and discussion in literature was added. We present here the paragraphs with large changes:

"In addition to the short-term ozone recovery, during the analysed period was observed reductions as consequence of the ozone hole influence. The ground-based SAOZ and satellite OMI instruments reflected maximum reduction of around 30% in TOC. Similar reduction has been found in Wolfram et al. (2012) during November 2009, while Kirchhoff et al. (1997) had reported maximum reduction of around 60% by 1992-1994 at similar latitudes (Punta Arenas, Chile) respect the monthly mean values. If we analyse the ozone reduction in altitude, we observed maximum decreases of 20% and 25% respect the climatology value at 27 km and 37 km, respectively. DIAL measurements of ozone profiles carried out in the OAPA have shown maximum differences of around 50% in September-November (WMO, 2013; WMO, 2012; WMO, 2011b). These results highlight the importance of measurements at sub-polar regions." "The MWR-MLS inter-comparison at 27 km reveals a MBE of 5%, which is consistent with the value obtained Ohyama et al. (2016). Boyd et al. (2007) also carried out similar inter-comparisons between MLS and two MWR installed in Mauna Loa, Hawaii and Lauder, New Zealand. The differences reported for Lauder range from +7% to 10% between ∼20 to ∼28km, while for Mauna Loa differences are around ∼3% (Figure 1, Boyd et al. (2007)). On the other hand, Fiorucci et al. (2013) reported a difference of 10% at 26 km of altitude. Thus, the comparisons carried out between MWR and MLS reveal good agreement for the considered altitudes, consistent with the results of other authors. Similarly, we analysed the MWR-DIAL comparison at 27 km and we can observe that the correlation coefficient (R = 0.73) and the MBE (1%) are consistent with those obtained by a similar inter-comparison carried out by other authors. Nagahama et al. (1999) obtained a

correlation coefficient of 0.77 and a MBE=1%, although that analysis was realized at 38 km. Studer's et al. results reflect a 1.43% of difference between MWR and DIAL comparison."

Conclusions

RC: What scientific progress was made in the study?

AC: The conclusion section was improved and the following progress are mentioned in the Conclusion section:

- The MWR ozone mixing ratio retrieved in Río Gallegos was compared for the first time with ground-based measurements from the ozone DIAL/NDACC instrument and satellite measurements from the MLS on board the AURA/NASA.

- As an example of MWR capability and use, this work focuses on an atypical event of the incursion of polar vortex and ozone hole influence over Río Gallegos, detected from the MWR measurements at 27 km and 37 km during November 2014. The event is then analyzed by the use of Advected Potential vorticity and ground-based and satellite measurements.

- The time series of the ozone mixing ratio with a temporal resolution of $\sim$1 hour from the Millimeter Wave Radiometer (MWR) installed in OAPA, Río Gallegos (51.6° S; 69.3° W) at different altitudes are reported for the first time. Río Gallegos is located in sub-polar latitudes, which makes it a suitable site to study stratospheric and mesospheric ozone due to its closeness to the Antarctic ozone hole.

- It is highlighted the importance of these measurements due to the lack of ground-based radiometer observations of ozone between Antarctic latitudes and mid-latitudes, allowing to improve the understanding of the stratospheric and low-mesospheric dynamic using the ozone mixing ratio as a tracer and improving the characterization of the dynamical models.

RC: As tip I suggest to merge what is written in the "Discussion".

AC: The Discussion section was improved considerably as we described above, with the intention to keep this section with discussion in literature as the referee proposed.

References RC: References - Put in alphabetical order.

AC: Reference was organized in alphabetical order

Figures: RC: Figure 2.2. Explain in the text why MWR fall data between March and April and July and August.

AC: It was included in the caption of the figure 2.

RC: Figure 5.1 should be in the methodology

AC: The figure was adapted to Material and Methodology section.

Please also note the supplement to this comment: https://www.ann-geophys-discuss.net/angeo-2019-17/angeo-2019-17-RC1supplement.pdf

(A more clear .pdf file is upload as supplement file)

Please also note the supplement to this comment:
https://www.ann-geophys-discuss.net/angeo-2019-17/angeo-2019-17-AC1-supplement.pdf
* * *
Ajuste Lineal:
y = 0.283*x + 6.01

Fig. 1.

|          | Alt.  | N  | Slope | Intercept [vmr(ppm)] | R    | MBE  |
|----------|-------|----|-------|----------------------|------|------|
| MWR-MLS  | 27 km | 84 | 1.01  | 0.24                 | 0.65 | +5%  |
|          | 37 km | 84 | 0.96  | -0.43                | 0.63 | -11% |
|          | 65 km | 84 | 0.95  | 0.02                 | 0.88 | -7%  |
| MWR-DIAL | 27 km | 30 | 0.93  | 0.36                 | 0.73 | -1%  |

**Fig. 2.**

---

## Author Comment (AC2) · 25 Apr 2019

The authors acknowledge the anonymous referee and the editor for the time spent to review this manuscript and also for their constructive comments. The manuscript was revised and improving according to the referee comments and suggestions.

(A clearer version is upload as Supplement file) Answers to the referee's comments.

Specific comments: - The manuscript's title must be as concise and direct as possible, emphasizing the object of study instead one of the used tools. I suggest some like: "Analysis of a November 2014 southern sub-polar short-term ozone variation event".

Eventually, if the MWR instrument is cited, please change "Millimiter" by "Millimeter".

AC: We find appropriated the title proposed by the Referee and we make some changes with the intention to include the instrument that allow the "short-term" study over sub-polar regions. The title was modified as follow:

"Analysis of a southern sub-polar short-term ozone variation event using a Millimeter-Wave Radiometer"

RC: A conceptual aspect to revise throughout the manuscript is the coherence and rigor in the use of terms "polar vortex" and "ozone hole". The "Antarctic polar vortex" is a dynamical phenomenon which has been present probably for millions of years, and their mention is essential when the dynamics is analyzed particularly as a function of the altitude. While, the "Antarctic ozone hole" is the extreme manifestation of the stratospheric ozone layer depletion in the interior of the "Antarctic polar vortex", which has made evident since late 1970s, and is mainly referred to either when their vertical ozone structure is afforded or their consequences on surface are analysed. To speak of "ozone hole", for deﬡnition the vertical total ozone column values must fall below 220 DU; authors must revise their use when appropriate. In turn, terms as "ozone hole influence" are appropriate for sub-polar regions but in this case explicit mention to the "Antarctic ozone hole" must be made, eventually an abbreviation AOH may be useful. Similarly, phrases as (page 12, lines 19-20) "the southern part of South America has been affected by the systematic and abrupt intrusion of the polar vortex during the spring since the 1980's" are inappropriate: as said, the Antarctic polar vortex occurs probably since millions years ago, the difference is that before the 1980s their interior produced no "ozone hole", i.e. ozone values below 220 DU as it is deﬡned, and without the presence of the ozone hole probably the polar vortex intrusions would have no major transcendence for the surface. Authors must take particular care about the use of these key expressions. In this phrase, also the word "systematic" is inappropriate. It could be changed by some like: "the southern part of South America has been affected by the frequent abrupt intrusions of the AOH during the spring since the 1980's".
Similarly, the phrase (page 11, line 14) "This decrease is related to the passage of the ozone hole over Rio Gallegos" is wrong, as TOC never falls below 220 DU. Several other paragraphs along the manuscript must be revised accordingly.

AC: The underlined sentences by the reviewer are now changed in the manuscript for better clarification and distinction between AOH influences and AOH overpass. The introduction was modified to clarify the terms used in the paper as "Ozone hole influence" and "ozone hole overpass". Please, see pg. 3, lines 9-13.

- Given that the vertical total ozone column (TOC) values are a necessary reference when ozone anomalies are reported, I suggest a detailed mention to the TOC not only when the present case is analysed but also when mention to other cited cases to help distinguish Antarctic ozone hole "influences" from Antarctic ozone hole "overpass", and ozone hole "reductions" from eventual "mini ozone-holes" or real ozone hole "overpass".

AC: As we mentioned above, the introduction was modified to clarify the terms used in the paper as "Ozone hole influence" and "ozone hole overpass". Please, see pg. 3, lines 9-13.

- In the Introduction: as a benchmark for the specific analysis of this work, it would have been desirable a characterization, based on references, of the known spring-time typical vertical structure of the atmosphere over southern South America on both "sides" (inner/outer) of the Antarctic ozone hole.

AC: The following paragraph has been added to put in context the ozone reduction due to the ozone hole influence or the ozone hole passing over the southern South America (pg. 2, line 19):

"In spite of the fact that massive ozone depletion is produced over the South Pole, the total ozone column (TOC) and vertical reduction were also observed in non-polar regions between 1980s and 1990s (WMO, 2014). Kirchhoff et al. (1997; 1997b) reported

TOC ranging from 145 DU to 250 DU in Punta Arenas (53.0'S, 70.9'W)), during low-ozone events during September-December of 1992-1995, for which the climatological average is 330-334 DU (maximum reduction of ∼56%). More recent study reported reduction of 40-45% in TOC over Río Gallegos (Kuttippurath et al. 2010b). ECC (Electro Chemical Cell) ozone-sonde profiles measurements reflect reductions of around 30 to 50% between 15 km to 32 km of altitude in ozone hole condition (inner) respect normal condition (outer the ozone hole) in Punta Arenas (Kirchhoff et al.,1997). Similar reduction was observed from a Differential Absorption Lidar at Río Gallegos (Wolfram et al., 2006)."

- In the same sense, specific parts of these references could be useful to compare and put in major context the results from this work.

AC: Discussion section were improved and references were added to put our results in context with other results. Please, see section 4. Discussion (pg. 11) for more details.

- Given that one of the concerns with ozone negative anomalies is the potential increase in harmful UVB solar irradiance at ground, please could you add, e.g. in Figure 6, other plot of locally-measured clear-sky UV Index (at noon, or at a given fixed solar zenith angle) allowing quantify the simultaneous UVB increase for these days?.

AC: Taking into account this comment, we decided to modify the figure 7 (ex figure 6) adding the daily maximum UVI measured from a ground-based instrument (Radiometer YES UVB-1). This instrument was installed in Río Gallegos by 2014 and it is part of the SAVER-NET radiation network (http://data.savernet-satreps.org/). We decided to present the daily maximum UVI due to the fact that most of the analysed days were partially cloudy with broken clouds, and the maximum UVI were measured near to the noon. The daily maximum UVI was added in Figure 7 as follow:

AC: In addition, a paragraph about the description of the daily maximum UVI was added in 3.2.1 Description of the case study section (pg. 11, line 13).

Minor comments:

Text - Please defi̧ne the abbreviations the fi̧rst time the parameters are mentioned, and then use just the abbreviation. E.g. page 11, line 14: standard deviation is mentioned before, abbreviation SD should be presented the fi̧rst time it is mentioned and then only SD used. The same for TOC in line 18.

AC: It was revised.

- Page 3, lines 1-2: please change by "due mainly to tropospheric-stratospheric dynamical processes".

AC: It was changed.

- Page 3, lines 6-10: I think a change in the order of paragraphs would make more coherent this sentence. I suggest: "The transport of polar air masses may take the form of "fi̧laments" and "tongue", which induce anomalies on the ozone and UV observations over mid-latitudes. Recently, based on satellite and ground-based observations in Uruguay and Southern Brazil, Bresciani et al. (2018) showed a decrease of ozone over these sites during October 2016 in link to this phenomenon".

AC: Taking into account this comment, the paragraph was changed as following (pg.3, line 19):

"The air-mass transport in the stratosphere has been extensively analysed using the advected potential vorticity (APV) which is considered a suitable dynamical tracer in the stratosphere. The transport of polar air masses may take the form of "filaments" or "tongue". These terms have been used to explain the transport of air from the edge of the polar vortex into middle latitudes by Waugh (1993) analysing potential vorticity maps, and previously by Randal et al. (1993), to explain the intrusion of tropical air into mid-latitudes. When the intrusion of air from the polar vortex reaches mid-latitudes and produce ozone decreases, it induces anomalies on the surface UV radiation. Bittencourt et al. (2018) also linked the occurrence of this event over South America to later

changes in the tropospheric and stratospheric dynamic behaviour. Thus, this parameter can be used to study the dynamics of the Antarctic polar vortex and as a tracer of poor-ozone air masses that are released from the ozone hole (Bittencourt et al., 2018; Kirchhoff et al., 1996, Pinheiro et al., 2011; Wolfram et al., 2012; Hauchecorne et al., 2002; Marchand et al., 2005; Bencherif et al., 2007)."

- Page 3, line 9: "which induce anomalies on the ozone and UV observations". Anomalies are on the ozone and UV behavior, not on the observations. Please correct.

AC: "UV observations" was change by "surface UV radiation" (pg. 3, line 24).

- Page 3, lines 22-23: phrase "The OAPA is located in sub-polar latitudes, which makes it a suitable site to study stratospheric ozone due to its closeness to the Antarctic ozone hole" is wrong. It could be: "The geographical location of OAPA makes it a suitable site to study the sub-polar stratospheric ozone due to its closeness to Antarctica".

AC: It was changed as suggested (pg.4, line 15).

- Page 3, line 31: "decreasing the ozone amount" instead "increasing the ozone amount"?.

AC: During the analysed period, decrease and local increase of ozone amount were observed at both altitudes (27 km and 37 km). In this part of the sentence we refer to the local increase, when the ozone mixing ratio at both altitudes (27 km and 37 km) present a local peak.

With the intention to clarify this point in the text, the sentence into the manuscript was changed as follow (pg. 3, line 30):

"The high temporal resolution (one hour) of the MWR observations are analysed at different altitudes (27km and 37 km) with the aim to determine the short-term variability of ozone mixing ratio and the moment when the polar vortex and its edge (as filamentary structure) with poor-ozone air masses pass over Río Gallegos and leave it at those altitudes, resulting in a local peak of ozone mixing ratio for a very short period of time

on November 2014."

- Page 4, some paragraphs of lines 1 up to 8 seem more appropriate for section 2. Materials and Methodology, other for the conclusions and future possibilities. Please redistribute them.

AC: The paragraph was modified and moved to the 2.1 Observations section. (Pg. 4, line 23)

The end of the paragraph was moved to the end of 5. Conclusion section. (Pg. 15 line 20)

- Page 7, line 11: define AMF.

AC: It was replaced by "air mass factor" (pg. 7 line 25)

- Page 8, line 3: "into the daily cycle": did you mean "within the diurnal cycle"?. In line 4: please rewrite "that this gas suffer in this layer" in other form.

AC: - "into the daily cycle" was change by "within the diurnal cycle" (Pg. 8, line 17). - The sentence was changed by "We observe a marked difference of ozone mixing ratio between day and night measurements due to the ozone photochemistry around this altitude" (Pg. 8 line 17-18:)

- Page 9, line 22: replace "Argentina" by "South America".

AC: It was replaced (Pg. 10, line 10)

Figures

- Text of Page 11, line 10: . . . "light red". . . but in the caption of Figure 6 it is referred to as "pink".

AC: The color was unified as light red.

- Figure 9 and several paragraphs from the Introduction treating on the characteristics of the measurement site (e.g. page 3, lines 21 on) should be at the start of section 2.

Materials and Methodology.

AC:

- Figure 9 (now figure 3) was moved to section 2 (Materials and Methodology) (pg.8, line 25).

- The paragraph describing the measurements site was moved to subsection 2.1 (Observation) and it was adapted as follow (Pg. 4, line 14):

"Ground-based instruments used here are operated at OAPA, Río Gallegos, Argentina (51.5° S; 69.3°W), belonging to CEILAP (hereafter OAPA). The geographical location of OAPA makes it a suitable site to study the sub-polar stratospheric ozone due to its closeness to Antarctica. Since 2005, a Differential Absorption Lidar (DIAL) has been operated at the OAPA with the aim to retrieve stratospheric ozone profiles (Wolfram, 2006; Salvador, 2011), which were joined to the Network for the Detection Composition Change (NDACC) in 2008 (http://www.ndsc.ncep.noaa.gov). In addition, a ground-based SAOZ spectrometer instrument (Pommereau and Goutail, 1988) to retrieve TOC was installed in the OAPA by 2008 and belongs to LATMOS/CNRS. To contribute to ozone monitoring, the Solar Terrestrial Environment Laboratory, Nagoya University, Japan, installed the MWR in OAPA in 2011, which incremented the temporal resolution and increased the altitude range of the ozone profiles (Orte et al., 2011; Orte 2017). On the other hand, satellite OMI and MLS datasets are used here to inter-compare with mentioned ground based."

- The captions of the figures must contain all the information needed to interpret them. Please revise the captions of all figures. In Figure 2 please correct . . . ratio for three altitudes: 27, 37 and 65 km.

AC: The captions of all figures were revised and improving. The caption of the figure 2 was corrected.

- The abscissas and ordinates legends and labels must explicit clearly the parameters

in each axis. E.g. in Figures 2, 4 and 5, the y-legends must include "ozone mixing ratio".

AC: "Ozone mixing ratio" legend was included in the y-legends of the mentioned figure (now Figure 2, 5 and 6). Also was included in Figure 7 (right axis).

Dates in Figure 3 are better presented in Figure 2.

AC: Date of the mentioned figure was modified.

In Figures 4, 5 the altitude may be in form of title for each plot. In Figure 6 the year is not specified, don't use the abbreviation TOC. AC:

- The altitude was moved as a title those (Figures now 5 and 6). - The year was specified at the bottom of the figure (now figure 7) and TOC was changed by "total ozone column"

These comments may be considered as relatively "minor changes". However, I suggest they should be taken as mandatory for a posterior re-evaluation of the manuscript.

Please also note the supplement to this comment:
https://www.ann-geophys-discuss.net/angeo-2019-17/angeo-2019-17-AC2-supplement.pdf

———————————————————

[Figure]

Figure 7. Atypical event of Antarctic ozone hole influence over Rio Gallegos. (a) (Bottom) Time evolution of the MWR ozone mixing ratio at 27 km (red line) and 37 km (blue lines). Light red and light blue areas represent the ozone mixing ratio zonal climatology at both altitudes calculated using MLS database (2004 - 2016). (Top) Time evolution of total ozone column measured with the ground-based SAOZ instruments (green dots) and OMI (purple dots) in Dobson Units. White line and grey area represent the climatology and one SD calculated using the OMI data-base (2004 - 2017). (b) Time Evolution of the Daily maximum Ultraviolet Index measured with the ground-based solar radiometer YES UVB-1 at OAPA. White line and grey area represent the climatological UVI at noon in Rio Gallegos.

**Fig. 1.**

---

## Referee Report (RR1)

**Orte et al., 2019:**

**"Analysis of a southern sub-polar short-term ozone variation event using a Millimeter-Wave Radiometer".**

**General comments:**

The authors present a study about an atypical event of polar vortex and ozone hole influence over Río Gallegos during November of 2014. This event was detected from the Millimeter Wave Radiometer (MWR) measurements at 27 and 37 km and the advected potential vorticity (APV) was calculated from the high-resolution advection model MIMOSA (Modélisation Isentrope du transport Mésoéchelle de l'Ozone Stratosphérique par Advection) at 675 and 950 K to understand and explain the atmospheric dynamic related to ozone rapid variation during the passage of the polar vortex. In addition, the MWR dataset were compared for first time with Microwave Limb Sounder (MLS) to 27 km, 37 km and 65 km and with the Differential Absorption Lidar (DIAL) installed in Observatorio Atmosférico de la Patagonia Austral (OAPA) between October 2014 and 2015.

This work is a useful representation of the important contribution made by the Millimeter Wave Radiometer (MWR) at Río Gallegos and certainly, understand the ozone hole influence over Río Gallegos is of fundamental importance in many environmental processes which can lead to increases in the UV radiation on the surface. This increase in the UV radiation related to ozone reductions can be dangerous to life on earth and it represents a significant scientific advance.

After the first review, major corrections were suggested and in general the authors were able to remedy the main failures observed. The Point-by-point response to the referee comment's was very clear and precise in most points, however small details can still be improved especially in discussing results in the literature, although this has already been greatly improved.

Because of these I would recommend to accept with Minor Revisions this manuscript. Also, because I am not a English language native speaker, I suggest to the editor to check if the English is proper for publication.

**Specific comments:**

- ➢ In the **abstract:** Requested modifications are made.

- ➢ In the **1. Introduction:** Requested modifications are ok.

- ➢ In the **2. Materials and methods**: Requested modifications are ok. However, a subsection should be created describing the UV radiation data used in this new version of the manuscript.

- 2.3
  Why the comparison of the MWR with DIAL occurs only for the 27 km height. It could also occur for 37 km. Or is there any impediment?

➢ In the **3. Results:** Requested modifications are ok.

- Y axis of figure 7a (top) should contain values throughout the graph and contemplate values close to 230 DU observed.

➢ In the **4. Discussion**

Creating a discussion section instead of discussing the results as they occur in the text is always a dilemma. You can make the mistake of not discussing some results in the literature. Comparing the results with what was discussed. It is observed that not all results were properly discussed as listed below:

- In Pg 11, line 5: "The difference between measurements can be attributed to the typical uncertainties of each instrument, although another source of difference is introduced due to the non-collocated measurements inter-compared". This affirmation was discussed but not referenced.

- Discussion on results related to UV radiation should be attached.

- Discussion on results related to AOH influences should be improved, mainly in relation to the advection of potential vorticity process which caused the observed ozone reduction.

- The first paragraph of the discussion has affirmations without reference.

- The present affirmation was not discussed: "When we compare the MWR with the MLS, it is considered that both instruments are measuring the same air masses, although the location of the satellite measurements differs from the location of the MWR measurements, which can introduce a difference in the ozone mixing ratio measured."

- The present affirmation was not discussed: "One reason why the correspondence between the MWR and the DIAL is greater with respect to the MLS may be that the two instruments installed on the ground (MWR and DIAL) are monitoring the same air mass, while the distance with the location of the MLS observations could be introducing differences in the comparison".

- The present affirmation was not discussed: "It is important to note that the MWR and DIAL instruments retrieve ozone in different fundamental units. While the MWR 30 provides the ozone mixing ratio, the DIAL provides the ozone number density as a function of altitude. The DIAL unit was converted to the MWR unit for the inter-comparison using the temperature and pressure retrieved from the DIAL. Thus, uncertainties in these parameters could be adding uncertainties in the ozone amount in ppm from the DIAL".

➢ **Conclusions**
  - Requested modifications are ok.

---

## Author Response (AR2)

**Author comment's**

The authors acknowledge to referees and the editor for the time spent to review this manuscript and also for their constructive comments.

**Point-by-point response to the referee comment's 1: Lucas Vaz Peres**

The authors acknowledge to Prof. Lucas Vaz Peres for the time spent to review this manuscript and also for their constructive comments.

The manuscript was revised and improving according the referee comments and suggestions.

The specific answers are **in blue**, while the referee comments are **in black.**
* * *
 "Analysis of a southern sub-polar short-term ozone variation event using a Millimeter-Wave Radiometer".

General comments:

The authors present a study about an atypical event of polar vortex and ozone hole influence over Río Gallegos during November of 2014. This event was detected from the Millimeter Wave Radiometer (MWR) measurements at 27 and 37 km and the advected potential vorticity (APV) was calculated from the high-resolution advection model MIMOSA (Modélisation Isentrope du transport Mésoéchelle de l'Ozone Stratosphérique par Advection) at 675 and 950 K to understand and explain the atmospheric dynamic related to ozone rapid variation during the passage of the polar vortex. In addition, the MWR dataset were compared for first time with Microwave Limb Sounder (MLS) to 27 km, 37 km and 65 km and with the Differential Absorption Lidar (DIAL) installed in Observatorio Atmosférico de la Patagonia Austral (OAPA) between October 2014 and 2015.
This work is a useful representation of the important contribution made by the Millimeter Wave Radiometer (MWR) at Río Gallegos and certainly, understand the ozone hole influence over Río Gallegos is of fundamental importance in many environmental processes which can lead to increases in the UV radiation on the surface. This increase in the UV radiation related to ozone reductions can be dangerous to life on earth and it represents a significant scientific advance.
After the first review, major corrections were suggested and in general the authors were able to remedy the main failures observed. The Point-by-point response to the referee comment's was very clear and precise in most points, however small details can still be improved especially in discussing results in the literature, although this has already been greatly improved.
Because of these I would recommend to accept with Minor Revisions this manuscript. Also, because I am not a English language native speaker, I suggest to the editor to check if the English is proper for publication.

Specific comments:

* In the abstract: Requested modifications are made.

* In the 1. Introduction: Requested modifications are ok.

\* In the 2. Materials and methods: Requested modifications are ok. However, a subsection should be created describing the UV radiation data used in this new version of the manuscript.

The following subsection was included (pg. 8, line 3):

**"2.1.5 Solar Radiometer YES UVB-1**

*The ground-based radiometer YES UVB-1 (Yankee Environmental System, Inc.) installed in OAPA is used to measure the erythemal irradiance UVB at surface, and the UVI is retrieved. It is connected to a data logger, which is configured to acquire one measurement per minute. Due to the UVI is strongly affected by the ozone amount in the atmosphere, the time evolution of the daily maximum UVI is analysed during the period of the case study. We decided to present the daily maximum UVI instead of the UVI at solar noon due to the fact that most of the analysed days during the low ozone event were partially cloudy and the maximum UVI were observed near the solar noon. Thus, the daily maximum UVI is more representative in terms of the low ozone amount impact over the clear sky UVI at surface."*

In addition, the following sentence was included in the "1 introduction" section and in the "2.1 Observation" section to introduce the UVI measurements, respectively:

*Page 4, line 15: "Finally, the solar Ultraviolet Index (UVI) at surface is also analysed during the event."*

*Page 4, line 28: "Due to the relationship between ozone amount and the solar UVB radiation at surface, this parameter is also measured in the OAPA with a ground-based solar radiometer YES UVB-1 (Yankee Environmental System, Inc.)."*

\* 2.3
Why the comparison of the MWR with DIAL occurs only for the 27 km height. It could also occur for 37 km. Or is there any impediment?

The power of the laser used in the DIAL installed in the OAPA was not stronger enough to reach 37 km. For this reason, the comparison between MWR and DIAL are not possible at this altitude.

\* In the 3. Results: Requested modifications are ok.

\* Y axis of figure 7a (top) should contain values throughout the graph and contemplate values close to 230 DU observed.

The value "240" was added in the Y axis (page 30).

\* In the 4. Discussion

creating a discussion section instead of discussing the results as they occur in the text is always a dilemma. You can make the mistake of not discussing some results in the literature. Comparing the results with what was discussed. It is observed that not all results were properly discussed as listed below:

- In Pg 11, line 5: "The difference between measurements can be attributed to the typical uncertainties of each instrument, although another source of difference is introduced due to

the non-collocated measurements inter-compared". This affirmation was discussed but not referenced.

To clarify this sentence, it was re-write as follow:

"The difference between measurements can be attributed to the typical uncertainties of each instrument, although another source of difference could be explain by the non-collocated measurements inter-compared"

This sentence is followed by the "This point is discussed in section 5". Into the discussion section were included the reference Sugita et al. (2017). The following sentence references the suggestion in *Discussion* (Page 15, line 13):

 *"Comparisons between DIAL and MLS were realized by Sugita et al. (2017) for an unusual case of persistence of the AOH over Río Gallegos occurred during November 2009, who also attributed part of the differences to the non-colocation of the measurements."*

- Discussion on results related to UV radiation should be attached.

The following paragraph is added (pg 14, line 15):

*"The time evolution of the daily maximum UVI was also analysed during the study period. As expected, we find an opposite behavior respect the total ozone column, which is in agreement with other results reported (Casiccia et al., 2008; Wolfram et al., 2012). It is observed a local minimum when the measurements of the ozone amount retrieved by the MWR presents a maximum at both altitudes."*

- Discussion on results related to AOH influences should be improved, mainly in relation to the advection of potential vorticity process which caused the observed ozone reduction.

The following paragraph is added (pg 13, line 30):

*"The influence of the polar vortex during the analysed period was confirmed in the APV from the MIMOSA model at two isentropic levels (675K and 950K). We observed filaments of air-mass from the polar vortex at both potential temperature levels passing over Río Gallegos. Similar cases of filaments travelling toward mid latitudes in the South Hemisphere have been reported analysing the APV (Waugh, 1993) without the possibility to report the stratospheric ozone amount with the time resolution reported here."*

- The first paragraph of the discussion has affirmations without reference.

Reference were added (pg 13, line 25):

*"It is well known that the southern part of South America is affected by the frequent abrupt intrusions of the AOH during the spring (Wolfram et al. (2012), Kirchhoff et al. (1997), WMO, 2013; WMO, 2012; WMO, 2011b)"*

- The present affirmation was not discussed: "When we compare the MWR with the MLS, it is considered that both instruments are measuring the same air masses, although the location of the satellite measurements differs from the location of the MWR measurements, which can introduce a difference in the ozone mixing ratio measured."

This sentence is a kind of introduction of the discussion that it does is discussed in the next two paragraphs. It is not an affirmation. It suggest that the difference between the MWR and MLS can be attributed to the difference in the location of the measurements under the

criterion of closeness between measurement described in the "2.3 Methodology and consideration".

This sentence can be separated in three parts:

1. "When we compare the MWR with the MLS, it is considered that both instruments are measuring the same air masses"
   We can to consider that both instruments are measure the same air mass under the criterion of closeness that is explain in the second paragraph of the "2.3 Methodology and consideration" section. Strictly, both measurements are not in the same location because the location of the satellite measurements differs from the location of the MWR measurements, which is expressed in the second part of the sentence.
2. "The location of the satellite measurements differs from the location of the MWR measurements"
   This part of the sentence was analized in the section "2.3 Methodology and consideration" and in Figure 3. It is a fact.
3. "which can introduce a difference in the ozone mixing ratio measured"
   In this part of the sentence we put the word "can" to emphasize the suggestion that the non-strictly-collocated measurements could introduce differences in the intercomparison, which is obvious.

In addition, a reference was included in the discussion of this affirmation in pg. 15, line 14 which suggest the same *("Comparisons between DIAL and MLS were realized by Sugita et al. (2017) for an unusual case of persistence of the AOH over Río Gallegos occurred during November 2009, who also attributed part of the differences to the non-colocation of the measurements.")*

To clarify the sentence, we decided to re-write it as follow (change the "it is" by "we" and change the word "can" by "could"):

> *"When we compare the MWR with the MLS, we considered that both instruments are measuring the same air masses, although strictly the location of the satellite measurements differs from the location of the MWR measurements, which could introduce a difference in the ozone mixing ratio measured."*

- The present affirmation was not discussed: "One reason why the correspondence between the MWR and the DIAL is greater with respect to the MLS may be that the two instruments installed on the ground (MWR and DIAL) are monitoring the same air mass, while the distance with the location of the MLS observations could be introducing differences in the comparison".

It is a suggestion at the same way than the previous comment.

Here, what we want to say is that as the ground based MWR and DIAL instruments are monitoring ozone in the same place, the comparison are expected (and it is) to be better than the comparison between the MWR and the MLS, which are monitoring the ozone in the different location. We suggest a possible reason about why the intercomparison between the collocated instruments (MWR and DIAL) is better than between non-strictly-collocated instruments (MWR and MLS).

- The present affirmation was not discussed: "It is important to note that the MWR and DIAL instruments retrieve ozone in different fundamental units. While the MWR provides the ozone mixing ratio, the DIAL provides the ozone number density as a function of altitude. The DIAL unit was converted to the MWR unit for the inter-comparison using the temperature and

pressure retrieved from the DIAL. Thus, uncertainties in these parameters could be adding uncertainties in the ozone amount in ppm from the DIAL".

This paragraph refer the error propagation which increment the uncertainties when we convert the measurement from the fundamental unit of the DIAL (molecules/cm3) to other unit (ozone mixing ratio) using other measurements (pressure and temperature for this case). When we convert the unit of any measurement to another using other measurements, there is error propagation intrinsic that come from the error of each measurement.

\* Conclusions
- Requested modifications are ok.

**Point-by-point response to the referee comment's 2:Anonymous.**

The authors acknowledge the anonymous referee for the time spent to review this manuscript and also for their constructive comments.

The manuscript was revised and improving according to the referee comments and suggestions.

The specific answers are in blue, while the referee comments are in black.
* * *
Revision of MS No.: angeo-2019-17-R1

Title: Analysis of a southern sub-polar short-term ozone variation event using a Millimeter-Wave Radiometer.

Authors: Pablo Facundo Orte, Elian Wolfram, Jacobo Salvador, Akira Mizuno, Nelson Bègue, Hassan Bencherif, Juan Lucas Bali, Raúl D'Elia, Andrea Pazmiño, Sophie Godin-Beekmann, Hirofumi Ohyama, Jonathan Quiroga.

Overall evaluation:

The manuscript has definitely improved. However, this second revision evidences still several necessary corrections. Again, these comments may be considered as relatively "minor changes", but in my opinion they are mandatories to consider the manuscript as acceptable for publication:

Specific comments:

- Abstract: sentence "The measurement shows a very short-term recovery in the middle of ozone mixing ratio decrease that could be detected by the MWR" is confuse. Please clarify.

To clarify the sentence, it was replaced by the following sentences:

*"During the event, the MWR observations at both altitudes show a decrease of ozone followed by a local peak of ozone amount of the order of hours. This local recovery is observed thanks to the high time resolution of the MWR mentioned."*

- The concepts of potential temperature, isentropic surface, isentropic level, are introduced after temperatures 675 K and 950 K are mentioned, starting with the Abstract. Please define the concept as at the beginning as possible and unify terminology. In my opinion, the denomination of "isentropic level" is the most appropriate and should be used to avoid different denominations for the same concept.

The concept of isentropic level was introduced in the abstract in brackets:

*"The advected potential vorticity (APV) calculated from the high-resolution advection model MIMOSA (Modélisation Isentrope du transport Méso-échelle de l'Ozone Stratosphérique par Advection) was also analysed at two isentropic levels **(levels of constant potential temperature)** of 675 and 950 K (~27 km and ~37 km of altitude, respectively) …"*

The terminology was unified to "isentropic level":

- Page 8, line 14: "Isentropic surface" was changed by "isentropic level".
- Page 13, line 6 and 7: "Potential temperature" was changed by "isentropic level".
- Page 33 (figure caption of the figure 8): "Maps show the evolution of the polar vortex for two isentropic levels with potential temperatures of 675K (left) and 950K (right)."

- In the same aspect, the fact that isentropic levels 675 K and 950 K correspond to approximately ~27km and ~37km is explained too late in the text (page 12). Please clarify this fact as at the beginning as possible in the text.

It was clarified in the abstract, page 1, line 30 in brackets (please, see previous comment).

- Page 2, line 32: sentence "Ground and space-based observations and models have shown an increase of the total ozone since 2000" is incoherent with the two following sentences: "Nevertheless, this increase is not significant for the period 2000-2013 (WMO, 2014). Ball et al. (2018) extended this period from 1998 to 2016 and concluded that there are non-significant changes in the total amount of ozone from merged ozone datasets". Please clarify.

To clarify the sentence, it was removed and the following sentence (**in bold**) was added:

*"Together with the banning in the use of ODS set by the 1987 Montreal Protocol, the general expectation was that the TOC would recover as the amount of ODS decreased in all regions. Recent studies showed a recovery of the stratospheric ozone column during September (statistically significant) and October (statistically insignificant) for the South Polar Region (Salomon et al., 2016; Weber et al., 2018; Pazmiño et al., 2018). **Outside the polar region (between 60 S and 60 N) Ball et al. (2018) concluded that there are non-significant changes in the total amount of ozone between 1998-2016 from merged ozone datasets, although they reported a decrease in the stratospheric ozone layer, which this findings imply an increase in the tropospheric ozone."***

- Page 3, line 20: Bresciani et al. (2018) named this phenomenon "secondary effect of the Antarctic ozone hole", concluding that "Data revealed the poor ozone air mass trajectory from some days before arriving in southern Brazil and Uruguay to some days after its passage, and confirmed its polar origin", but… could this sentence be interpreted as the authors refers to?: an actual pass of the polar vortex over Uruguay and Southern Brazil?.

Here, we want to describe the influence of the AOH affecting the Uruguay and Southern Brazil region. The sentence was changed by (in bold):

*"This phenomenon was first observed by Kirchhoff et al. (1996) and reported by Pinheiro et al. (2011) in South America.* **Recently, based on satellite and ground-based observations in Uruguay and Southern Brazil, Bresciani et al. (2018) showed a decrease of ozone over these sites during October 2016 due to the influence of the AOH reaching mid-latitudes."**

- Page 11, line 13: R=0.68 is mentioned for altitude 27 km, but in item 3.1.1, table 1 and figure 5 it is referred to as R=0.65.

It was revised. The R=0.68 was changed by 0.65.

- Page 12, paragraph starting in line 15: an adaptation of the phrase from the answers to referee "We decided to present the daily maximum UVI due to the fact that most of the analysed days were partially cloudy with broken clouds, and the maximum UVI were measured near to the noon" must be included in the text of the manuscript.

The sentence in bold was included:

*"Figure 7b (blue dots) presents the time series of the daily Ultraviolet Index (UVI) maximum (near the* **solar** *noon) during the low ozone event described before measured with a radiometer YES UVB-1 (Yankee Environmental System, Inc.) installed in the OAPA.* **We decided to present the daily maximum UVI instead of the UVI at solar noon due to the fact that most of the analysed days were partially cloudy and the maximum UVI were observed near the solar noon and it are more representative in terms of the low ozone amount impact over the UVI."**

- Figure 8: plots on the left (675 K) and on the right (950 K) look too similar given that the same colour scale is used in both, but in fact the actual values of APV are totally different. I suggest unify the scale of colours using the same scale on both sides with the total range APV (0-900), but still highlighting with colours and tones the features you want emphasize: for instance the same present colours in the range 0-200 (blue to yellow) but then passing to (red to pink) in the range 200-900.

The colour scale of plot on the right (950K) was changing to avoid confusion between both scales. We took the suggestion of the referee and some colour scales were proved and we decided to set the colour as we present in the new version of the manuscript. We considered that this colour scale highlight the features that we want to show in the event.

Minor comments:

- Page 2, line 5: Add comma: "It acts as an absorber of harmful solar UVB radiation, …".

Comma was added.

- Page 2, line 7: please change "Although most production takes place in the equatorial region" by "Although most production takes place in the mesospheric-stratospheric equatorial region".

It was changed.

- Page 2, line 21: Please revise sentence "the total ozone column (TOC) and vertical reduction", it could be better simply "ozone reduction at different height levels".

It was changed.

- Page 2, line 25: "More recent study reported reduction of 40-45% in TOC over Río Gallegos", please specify the date of the reported TOC reduction.

It was specified as follow:

*"More recent study reported reduction of 40-45% in TOC over Río Gallegos on October 2008 and November 2009 (Kuttippurath et al. 2010b)."*

- Page 3, line 15: change by "The passage of the AOH is identified using the TOC threshold of 220 DU".

It was changed.

- Page 4, line 27: change by "and the unique installed in subpolar region".

It was changed.

- Page 4, line 31: Did you mean "improving the validation of the dynamical models"?.

The word was changed by "validation".

- Page 10, line 4: remove word "system": 3.1 Inter-comparison of MWR with DIAL and MLS observations.

I was removed.

- Page 10, line 10: change by "campaigns become".

It was changed.

- Page 12, line 27: change "at both potential temperatures" by "at both isentropic levels".

It was changed.

English must be still revised in many sentences throughout the text, only for example:

- Page 1, line 30: "explain the dynamics".

It was changed.

- Page 2, line 20: "and they will remain for decades".

It was changed.

- Page 2, line 25: "A more recent study reported a reduction of 40-45% in TOC over Río Gallegos".

It was changed.

- Page 2, line 27: "respect to normal".

It was changed.

- Page 3, line 26: "… reaches mid-latitudes and produces".

It was changed.

- Page 4, line 31: "at these latitudes".

It was changed.

- Page 11, line 26: Remove word "on": "during November 2014".

It was removed

- Page 12, line 15: change by "daily maximum Ultraviolet Index (UVI)".

It was changed.

- Page 13, line 15: remove "from": "suffers sudden".

It was removed